# Recent Developments of Quantum Dot Materials for High Speed and Ultrafast Lasers

**DOI:** 10.3390/nano12071058

**Published:** 2022-03-24

**Authors:** Zhonghui Yao, Cheng Jiang, Xu Wang, Hongmei Chen, Hongpei Wang, Liang Qin, Ziyang Zhang

**Affiliations:** 1School of Electronic and Information Engineering, Qingdao University, Qingdao 266071, China; zhyao2018@sinano.ac.cn (Z.Y.); cjiang2017@sinano.ac.cn (C.J.); hpwang2020@sinano.ac.cn (H.W.); 2School of Nano-Tech and Nano-Bionics, University of Science and Technology of China, Hefei 230026, China; xwang2017@sinano.ac.cn (X.W.); hmchen2015@sinano.ac.cn (H.C.); lqin2019@sinano.ac.cn (L.Q.); 3Qingdao Yichen Leishuo Technology Co., Ltd., Qingdao 266000, China

**Keywords:** molecular beam epitaxy, quantum dots, active electrical pumped lasers, semiconductor saturable absorber mirrors, mode-locked laser

## Abstract

Owing to their high integration and functionality, nanometer-scale optoelectronic devices based on III-V semiconductor materials are emerging as an enabling technology for fiber-optic communication applications. Semiconductor quantum dots (QDs) with the three-dimensional carrier confinement offer potential advantages to such optoelectronic devices in terms of high modulation bandwidth, low threshold current density, temperature insensitivity, reduced saturation fluence, and wavelength flexibility. In this paper, we review the development of the molecular beam epitaxial (MBE) growth methods, material properties, and device characteristics of semiconductor QDs. Two kinds of III-V QD-based lasers for optical communication are summarized: one is the active electrical pumped lasers, such as the Fabry–Perot lasers, the distributed feedback lasers, and the vertical cavity surface emitting lasers, and the other is the passive lasers and the instance of the semiconductor saturable absorber mirrors mode-locked lasers. By analyzing the pros and cons of the different QD lasers by their structures, mechanisms, and performance, the challenges that arise when using these devices for the applications of fiber-optic communication have been presented.

## 1. Introduction

Semiconductor lasers emitting at wavelengths within the O band (1260–1360 nm) and the C band (1530–1565 nm) have attracted much attention in optical communications due to the low chromatic dispersion and optical loss in standard optical fibers. The performance of semiconductor lasers essentially depends on the properties of the heterojunction structures, mainly including the active region and the cladding layer structure. In the early stage, bulk materials were used as the active region in semiconductor laser heterostructures, in which carriers are free in the three-dimensional space, leading to only a few injection carriers contributing to the lasing mode. Since the first investigation of the optical properties in quantum well (QW) structures by Dingle et al. [1], a new kind of ultra-thin QW layer, applicated as the main active medium in semiconductor lasers [2,3,4], has been extensively investigated due to the physical interest as well as its superior characteristics, such as lasing wavelength tunability, low threshold current density, high characteristic temperature (T_0_), and excellent dynamic properties compared to the bulk materials based lasers [5,6,7,8,9]. However, the T_0_ of the threshold current is usually lower than expected for 1.3 μm QW lasers because of the small energy gap between the optical transition energy of QWs and the bandgap of the cladding layers [10,11,12], which strictly restricts its applications in the high-temperature environment. To further improve the performance of laser devices, it is highly critical to limit the carriers in a smaller space and a more concentrated density of states. In 1982, quantum dot (QD) lasers were theoretically proposed by Arakawa and Sakaki [13]. Carriers in the QDs can be confined in all of three-dimensional space; energy levels become completely separated and the density of state changes to a delta function. The thermal distribution in the QD structure is much smaller than that in the bulk and QW structures, which improves the device characteristics, such as the expected infinite T_0_. However, limited by the epitaxial growth technology of QD at that time, the early studies on the QDs and the devices were mainly about theoretical methods [13,14,15]. By 1994, Hirayama et al. had successfully realized the first QD laser lasing via the etching and regrowth technique for QW structures [16]. However, the QD laser exhibited a threshold current density as high as 7.6 kA/cm^2^ at 77 K under pulse current injection, which is over an order of magnitude higher than the threshold of the standard QW laser when cutting out from the same wafer measured at room temperature (RT) under continuous wave (CW) operation. Thanks to the considerable strides in epitaxial growth techniques of QDs, the self-assembled QDs with high uniformity, high density, and, most importantly, with almost no dislocations in and around the QD ensemble could be successfully manufactured to avoid etching damage [17,18]. In 1994, Kirstaedter et al. reported the first self-assembled QD-based semiconductor laser, which showed the threshold current density of 120 A/cm^2^ at 77 K and the calculated T_0_ of 350 K at the temperature range of 50–120 K, much higher than the theoretical prediction of 285 K for the QW laser [19]. After that, many breakthroughs of QD lasers were springing up [20,21,22,23,24]. Though the appearance of self-assembled QDs has opened up a new way for near-infrared semiconductor lasers towards the optical communication applications, it is challenging to extend the emitting wavelength of self-assembled QDs towards the long-wavelength range due to the crystal quality degrading with greater strain and the phase separation. In 1998, Huffaker et al. reported the first RT lasing at 1.3 μm InGaAs/GaAs of a QD-based laser with a low threshold current density of 270 A/cm^2^ [20]. However, the thermal broadening of the holes in the valence band of the QDs, due to close energy levels, resulted in the lower T_0_ at RT, and the modulation bandwidth of 1.0–1.3 µm of the self-organized In(Ga)As QD lasers was limited to ∼6–8 GHz [25]. To address these issues, the techniques of modulation p-doping and tunneling injection of the QD structures were proposed [26,27,28,29], and the high differential quantum efficiency of 88% [30], the low RT-CW threshold current density of 17 A/cm^2^ [31], and the infinite characteristic temperature from 5–75 °C have been successfully achieved for 1.3 μm QD lasers [32]. Even under 220 ℃, the CW lasing operation could be achieved with more than 1 mW of optical power [33,34]. In addition, with the commercially produced 10 Gb/s 1.3 μm QD lasers a high operating temperature of up to 100 °C has been realized by QD Laser Inc. [35]. Following the introduction of a strained InGaAs cap layer for InAs/GaAs QDs to reach 1.3 μm, to realize a QD laser working at the 1.55 μm range would simultaneously require higher indium content capping layers with a higher indium content and larger size InAs/GaAs QDs [36,37]. However, the crystal and optical properties will be rapidly degraded due to the high indium concentration in QDs and their surrounding matrix, leading to a large number of non-radiative recombination centers. Another way to solve this issue is to utilize quaternary InGaNAs [38] or InGaAsSb [39] to cap InAs/GaAs QDs, but this complicates the epitaxial growth process. Besides, InAs/GaAs QDs grown on thick metamorphic InGaAs buffer layers/virtual substrates have been used to extend emission around 1.5 μm [40]. However, both the repeatability and the reliability are issues for such structures. Previously, Zhang et al. developed an asymmetric In(Ga)As/GaAs dot-in-well (DWELL) QD semiconductor saturable absorber mirrors (SESAM) structure and realized the first 10 GHz repetition rate QD-SESAM mode-locked laser operating at the 1.5 μm wavelength range [41]. An InP substrate has the advantage of realizing longer-wavelength InAs QDs due to the smaller lattice mismatch (3%), instead of GaAs substrates. Therefore, 1.55 μm InP-based quantum dash lasers were reported by many research groups [42,43,44]. The study of QDs is continuing on the epitaxy growth, the design, and the fabrication to improve the performance of the QD devices [45,46]. The review is organized as follows: in Section 2, two main growth methods of QD structures, including top-down and bottom-up approaches are briefly described; in Section 3, the physics and device performance of QDs, including the modal gain, linewidth enhancement factor, temperature dependent optical properties, carrier dynamics of QD materials, low threshold current density, high temperature insensitivity, modulation characteristics, and high optical feedback tolerance of QD devices, are discussed; in Section 4, the active and passive QD-based laser devices in the context of fiber-optic communication, including Fabry–Perot (F–P) lasers, Distributed Feedback (DFB) lasers, Vertical-Cavity Surface-emitting lasers (VCSELs) and SESAM mode-locked lasers are reviewed. In addition, the prospects and challenges for the QD lasers are summarized at the end.

## 2. Growth Methods of QD Materials

QDs exhibit unique optoelectronic properties due to the strong quantum confinement of the carriers in three dimensions. However, it is a challenge to exploit the electronic properties of QD structures in optoelectronic devices, as many factors need to be realized simultaneously in the active region. The lateral dimension of the QD structures must be smaller than the de Broglie wavelength of the electrons in order to achieve the three-dimensional confinement of the carriers. For the structure of the QD laser especially, the injection process of the carriers and the proper optical waveguide structure must be considered to optimize the performance of the devices. Another major challenge for the QD laser in achieving predicted performances is the uniformity of the QDs. Here, there are two main techniques for the preparation of QDs to be reviewed, including top-down approaches using etching and regrowth processes and bottom-up approaches using self-assembled technology.

### 2.1. Top-Down Approaches

The fabrication of QD structures by patterning QWs was considered as a straightforward way in the late 1980s. The size, shape, and arrangement are controllable by this method for the realization of QD structures [16,47]. As shown in Figure 1, the fabrication processes usually include high-resolution lithography (e.g., electron beam lithography), dry/wet etching and regrowth. Etching is a popular method for realizing pattern arrays with high uniformity; however, it also brings a lot of nanometer-scale damage. It was highlighted by Clausen et al., who characterized dry-etched structures by using low-temperature cathodoluminescence (CL) [48]. The luminescence efficiency was found to be degraded with both the decreasing of the size of the nanostructure and the increasing of the etch depth, as shown in Figure 2. The thickness of the non-radiative surface layer increases with etching time, and the boundary of the damage layer moves towards the center of the structure, leading to where the smallest nanostructure can still emit light with the increase in the size. To overcome the etching damage and the planarization in the regrowth process of the etched features, significant efforts have been made in the fabrication technique. In 1994, the first QD laser was successfully achieved by Hirayama et al. using the top-down approach [16], in which the QD structure was fabricated by two-step metal-organic chemical vapor deposition (MOCVD) growth, electron-beam-exposure direct writing and wet-chemical etching, and the laser lasing at 77 K via excited states (ES) of the QDs due to the very low ground state (GS) gain of the QDs.

### 2.2. Bottom-Up Approaches

#### 2.2.1. Selective Growth Technology

An alternative method for preparing QDs with precise position and size control relies on the substrate pattern followed by the epitaxial growth [49,50,51,52,53]. As shown in Figure 3a, the tetrahedral shaped recesses were formed on a (111) B GaAs substrate by employing the SiO_2_ mask and selective chemical etch method and a size less than 20 nm at the bottom of the aperture [49]. A QD structure was formed at the bottom of the aperture using anisotropic growth on the facets. The photoluminescence (PL) and CL spectra of the QD structures formed at the bottom of the aperture are shown in Figure 3b. In 1998, Ishida et al. reported another bottom-up approach for the fabrication of QD structures [50], in which the GaAs QDs were selectively grown at the bottom of two-dimensional V-groove (2DVG) structures by using a SiO_2_ patterned substrate, as shown in Figure 4a. In Figure 4b, four stacks of GaAs QD structures with a lateral size of 40 nm were realized using this technique. However, the selective growth technology is not suitable for the fabrication of the laser devices because of the high loss in the waveguide structure. Another serious issue is the low modal gain due to a low areal dot density of only around of 10^8^ cm^−2^.

#### 2.2.2. Self-Assembled QD Technology

In principle, three typical growth modes of semiconductor heterostructures can be classified, governed by the lattice mismatch and the interfacial free-energy terms [17]. In lattice-mismatched material systems, the growth mode is driven by the sum of the interface energy (σ_i_), the epilayer surface energy (σ_e_) and substrate surface energy (σ_s_), Δσ = σ_i_ + σ_e_ − σ_s._ Δσ < 0, the layer-by-layer growth mode (Frank–van der Merwe, F–M mode), Δσ > 0, and the island growth mode (Volmer–Weber, V–W mode). The layer-then-island growth mode (Stranski–Krastanow, S–K mode) is a hybrid of the F–M and V–W growth modes, as shown in Figure 5. For the moderate lattice mismatch (7%), as shown in Figure 6a, the InAs QDs are grown through the S–K mode on the GaAs substrate. The InAs was deposited on the GaAs buffer layer with a layer-by-layer 2D mechanism at the initial stage, and the 2D thin layer was called the wetting layer (WL). With the increasing of the thickness of InAs, the strain is accumulated due to the lattice mismatch and small islands are gradually formed when the amount of the deposited InAs exceeds the critical coverage, which is usually around 1.6 monolayer (ML) (~0.5 nm) for the InAs deposited on the (001) GaAs substrate [54]. These small islands continue to grow with the addition of epitaxial materials until fully ripened. The driving force for the transition from 2D to 3D is the relaxation of the elastic energy with the increasing thickness of the mismatched epilayers. As shown in Figure 6b, the lattice parameter can be continuously relaxing along the growth direction, thus allowing the elastic energy to relax without misfit dislocations at the interfaces of the QDs. In the S–K growth process of the self-assembled QD, two aspects need to be considered: the processes of dynamics and thermodynamics [55]. Figure 7a shows the main dynamics process during the epitaxial growth, including adsorption, surface migration, nucleation, diffusion to the substrate (intermixing), step incorporation, and island incorporation. There are two main factors that determine the dynamic process. One is the substrate surface conditions before the QD epitaxial growth, including the morphology of the surface, orientation of the substrate, stress distribution, and any compositional gradient across the surface of the substrate. The surface density, as well as the diffusion of the adsorbed atoms, is another crucial aspect. These are major factors in the epitaxial growth process of QDs controlled by changing the growth conditions, such as the deposition rate and the substrate temperature. The study on the thermodynamic processes is also critical for the QD growth due to the high growth temperature and the low growth rate. The ad-atoms have enough time to migrate to their energy-balanced location, making the system close to thermodynamic equilibrium [56]. Figure 7b shows the changes in the total energy of lattice mismatched system versus time. Three stages for the S–K growth mode can be clearly observed. Phase A is 2D deposition, phase B is 2D–3D transition, and phase C is the ripening of the islands. In phase A, the epitaxial growth follows a 2D growth mechanism in the initial stage of the deposition; the accumulated elastic strain energy increases linearly with the deposited volume, resulting in a WL on the substrate. When the deposition time reaches t_cw_, the sTable 2D growth is changed to the 2D metastable growth, which means that the epilayers are ready to undergo a transition towards S–K morphology. Phase B can be divided into two steps: the nucleation and the growth. The nucleation activation energy is E_A_-E_E_, in which E_E_ is the excess energy in the metasTable 2D layer. When the islands occur, E_E_ decreases and the materials around the islands are consumed for the growth of the islands. There are two factors concentrated on the nucleation event in a fairly narrow period of time at the point X, as shown in Figure 7b. (1) The drop of “super saturation” and (2) the increase in the activation barrier for the thermally activated nucleation. The surface density of the QDs is determined by the concentration of nucleation points. With the formation of the islands, the strain has undergone local changes. The strain is high at the edge of the islands, and the larger island has the higher energy barrier. As long as the atoms are deposited on the surface of the WL, the new absorbed atoms will be incorporated into the formed islands, and the energy barriers of these islands will increase. Further capturing new atoms becomes more difficult. So, it has a size self-limiting effect on the QDs. Stage C represents the ripeness procedure; the growth process has lost most of the excess energy. There are still some slow interactions (e.g., diffusion of the materials on the surface) between the smaller and larger islands. Finally, the system will reach equilibrium. Among the many fabrication methods of QDs, self-assembled S–K growth technology is the most mature method for growing QDs with high density as well as high optical quality [57,58,59]. MBE and Metal-Organic Vapor Phase Epitaxy (MOVPE) are two sophisticated techniques widely used in compound semiconductor thin film growth. For the self-assembled QD under S–K growth mode, the most high-quality QD structures are based on MBE technology, due to the epitaxy structure with composition and doping profiles well controlled at the nanometer scale, while MOVPE is suited for the preparation of mass-production structures. In addition, it is worth mentioning that the strain-free GaAs/AlGaAs QD can be fabricated by the droplet epitaxy technique, which has a definite advantage in enabling a large number of high-quality QD layers as there are no strain induced dislocations [60,61]. The dot density and the dot size can also be precisely controlled by the droplet epitaxy technique on vicinal GaAs(111)A substrates, by which the fine structure splitting as low as 16 μeV was realized, thus making them suitable as photon sources in quantum communication networks [62]. Today, many QD devices have been commercialized, such as lasers, broadband light emitters, and passive devices, and among them, almost all are based on self-assembled QD structures.

## 3. Physics and Device Properties of QDs

Many remarkable properties of QD devices have essentially derived from the δ-function-like density of states, due to the three-dimensional confinement of the carriers. As shown in Figure 8a, the energy state density of the ideal QD material system is a set of discrete levels separated by regions of forbidden states, as the electron wave function is completely localized, arising from the same size, shape, composition, and uniformly distribution of the QDs in the parent material. Grundmann et al. reported ultra-narrow (0.15 meV) CL lines originating from single InAs QDs in a GaAs matrix for temperatures up to 50 K [63], directly proving their δ-function-like density of electronic states. However, the inhomogeneous distribution of QD size is essential, and the nonuniformity is more than 10% of the real self-assembled QDs [64,65,66], offering broad gain bandwidths, as shown in Figure 8b. Despite the fact that the inhomogeneous broadening cannot be avoided, QDs still exhibit unique photoelectric properties due to the strongly quantum confinement effect.

### 3.1. Physical Properties of QD Materials

#### 3.1.1. Modal Gain

Modal gain is vital for semiconductor lasers. For F–P lasers with the high modal gain, a high mirror loss can be tolerated, which in turn leads to higher slope efficiencies. Theoretically, the gain increases with increasing quantum confinement effects, as confirmed by Asada et al. [14]. However, the small areal coverage (~10%) of the plane of QDs and the inhomogeneity of the self-assembled process result in a small gain, suggesting that the modal gain of QD lasers is quite modest, with typical values of ∼4 cm^−1^ per QD layer. The saturable gain G_sat_ for QDs can be formularized as G_sat_∝N_QD_n_QD_Γ/Δ [67], where N_QD_ is the number of layers of QDs in the active region, n_QD_ is the surface density of the QDs in a single layer, Γ is the optical confinement factor, and Δ is the spectral broadening of the optical transition. Therefore, there are several ways to improve the gain by employing, either independently or in various combinations, the following: (1) stacking multiple QD layers in the active region to increase volume density (an increase in N_QD_) [68]; (2) the increase in the QD areal density in each layer using special growth conditions, such as dots in a well structure (an increase in n_QD_) [69]; (3) the decrease in QD size dispersion (a decrease in Δ) [70]; and (4) the use of high-contrast optical waveguides (an increase in Γ) [71]. Recently, the high modal of 54 cm^−1^ for long-wavelength QDs has been reported using nine stacked QD layer structures with high-density (8.0 × 10^10^ cm^−2^ per sheet) and high-uniformity (FWHM: 23 meV) [72]. A higher modal gain of 6 cm^−1^ per QD layer has been realized with seven stacked layers of QDs, in which GS lasing at 1.3 μm was achieved for a 360 μm cavity length at RT with as-cleaved facets [73]. To achieve a high modal gain, Maximov et al. tried to increase the optical confinement factor and dense stacking up to ten QD layers, realizing the modal gain as high as 46 cm^−1^ corresponding to the GS lasing at a 300 µm cavity length [71]. Recently, eight layers of high-density QDs have been successfully prepared by Tanaka et al., in which the large net maximum modal gain of 46 cm^−1^ was realized, and 25 Gb/s direct modulation was resultantly demonstrated for the first time in 1.3 μm QD lasers [74]. The reported values of the modal gain per layer in relation to the number of QD layers are generalized in Figure 9 [69,70,71,72,73,74,75,76,77].

#### 3.1.2. Temperature-Dependent Optical Properties

Temperature insensitivity is an obvious characteristic of QD laser superiority over QW lasers; so, it is necessary to investigate the temperature-dependent properties of QDs, especially for the applications in a long wavelength, because the lower confining potential and the smaller energy spacing between the GS and the first ES will reduce the thermal stability. As shown in Figure 10a,b, the temperature-dependent photoluminescence (PL) spectra of undoped as-grown QDs (QDU) and p-doped as-grown QDs (QDP) are measured at the temperature range from 4 to 300 K [78].

The PL linewidth, intensity, and peak positions are plotted with the temperature in Figure 11a–c. As shown in Figure 11a, the emission linewidth of the QDU sample decreases with the increasing of the temperature from 80 to 180 K, confirming that the carriers are redistributed among small and large QDs [79,80,81]. The emission line width rapidly increases when the temperature is above 180 K due to the increased electron-phonon scattering. For the QDP sample, the emission line width is wider than that of the QDU sample, which is mainly attributed to the state-filling effect induced by the p-doping modulation [82]. Figure 11b shows the normalized PL intensity as a function of the temperature for the QDU and QDP samples. The PL emission intensity of the QDU sample increases with the increasing of the temperature from 4 to 25 K. It is well known that the WL can work as a carrier reservoir, contributing to the increase in the PL emission intensity by an enhanced multiphonon-assisted relaxation process with the increasing temperature. At low temperature, the PL emission intensity shows no increase for the QDP sample, which indicates that the processes of carrier capturing and relaxation are dominated by the carrier–carrier scatterings due to the built-in holes [83]. With the temperature increase, the activated carriers escape from the GS to the states of nonradiative recombination centers or the GaAs barriers, resulting in a decrease in the PL intensity. Furthermore, it has been found that the PL intensity of the GS emission for QDP sample is less than that of the QDU sample in the temperature range of 4–200 K, which can be attributed to the increased density of the dopant-related trap states in the p-doped GaAs. Significantly, as shown in the inset of Figure 11b, the PL intensity of the QDP is still strong even at 300 K, which is about 16 times lower than that at 4 K, while there is a significant decrease of about 150 times for the QDU sample. For the QDP sample, the main reason for the slow quenching of the PL emission can be attributed to the higher GS gain, in which p-type modulation doping could compensate for the thermal escape of holes from the GS with the increasing of the temperature. Figure 11c shows the temperature dependence of the PL peak position from the QDU and QDP samples. It was found that the peak positions of the QDU and QDP samples all exhibit a drastic redshift with the increasing temperature. Combined with Figure 11a, the emission spectral width decreases minimally at 180 K, which suggests that the carriers transfer from the small QDs to the large QDs over WL at a higher temperature [79,84,85,86]. The PL peak energy position of the QDP sample appears smaller than that of the QDU sample over all the measured temperature ranges, due to the strong In/Ga element intermixing between the QDs and the surrounding barrier layers in the QDU sample originating in higher temperature growth process for the AlGaAs cladding layer. Additionally, for the QDP sample, modulation p-type doping can significantly reduce the Ga vacancy propagation, which leads to smaller In/Ga interdiffusion and a reduced intermixing effect [87,88,89]. For the temperatures ranging from 140 to 300 K, the peak position can be linearly fitted, and the slopes of the fitting lines are 0.35 and 0.29 meV/K, corresponding to the QDU and QDP samples, respectively. The smaller slope suggests that the QDP sample has a less temperature-dependent PL peak position.

#### 3.1.3. Carrier Dynamics

The dynamics of the carriers in QD structures is crucial from both the practical and the fundamental points of view. In general, carrier transfer from the barriers into the GS of the QDs can be divided into transport in the barriers, capture into the dots, and relaxation in the dots [90], as shown in Figure 12. The carrier capture and the relaxation rate from the barrier to the GS of the QD determine the fundamental limit for the laser modulation speed. The studies on carrier dynamics are usually carried out using ultra-fast optical techniques, such as pump probe spectroscopy and time-resolved PL (TRPL) [91,92,93,94,95,96,97].

##### Carrier Transport and Capture

There are two main mechanisms responsible for the carrier capture into the QDs, carrier–carrier (Auger) scatterings and phonon-assisted capture. For higher dimensional structures with a continuous state, such as QWs, the longitudinal optical (LO) phonon-assisted capture is the main mechanism due to the efficient energy dissipation. For QD structures, there are different mechanisms of carrier capture, depending on the differing sheet carrier density (n). For the phonon-assisted capture in QDs, the phonon-assisted capture rate is proportional to the sheet carrier density at low and moderate carrier densities. The single-phonon capture times were between 0.2 and 0.3 ps at a typical n- 10^17^ cm^−3^, and the two-phonon capture process is longer in the range of 1 to 10 ps [98]. Experimentally, Feldmanna et al. reported a fast capture process of ~10 ps with multiple phonon (4 LO + 1 longitudinal acoustic) emission [99]. On the other hand, for the carrier capture via Auger scatterings in QDs, the capture rate is very slow at low carrier densities, where the LO phonon-assisted capture is more efficient because only one carrier participates in the capture process. However, in the circumstances of high carrier concentrations, the capture rate of carrier–carrier scattering is proportional to n^2^ as two WL carriers are involved [100]; the capture rate of the Auger scatterings becomes similar to the phonon-assisted capture rate. The process of carrier dynamics can be studied by monitoring the PL rise times of different energies. Siegert et al. studied the PL rise times of different energies after excitation in the barriers at high excitation power and GS PL rise times for different excitation energies [101]. The WL rise times are close to ~2 ps for the undoped and p/n-doped samples, indicating that the modulation doping does not substantially modify the carrier transport in the barriers. The PL rise times for the QD transitions e4−h4 are 4.9, 5.4, and 6.1 ps for the p-doped, n-doped, and undoped samples. The difference in PL rise time between the QD e4−h4 transitions and the WL places the capture times at about 3–4 ps. To characterize the capture mechanism of the carriers, various excitation powers have been studied; the PL rise times do not experience any major dependence on the excitation power, increasing from 11.7 to 14.2 ps for the undoped sample and staying at around 5(6) ps for the p-(n-) doped samples when the average excitation power density is decreased from 230 to 1 W/cm^2^ (8 × 10^12^–4 × 10^10^ carriers/cm^2^). Such behavior is characteristic for the phonon-assisted capture.

##### Carrier Relaxation

The fast rate of relaxation of the carriers from ES to GS is beneficial for achieving high speeds in QD lasers. However, theoretical studies have suggested the existence of “phonon bottleneck” in QDs [102]. As the intersubband spacing of the dots is larger than the LO phonon energy, the GS and the ES are phonon decoupled. Thus, the intersubband relaxation time in QDs is expected to be much longer due to the suppressed phonon scatterings. However, other mechanisms have been suggested in QDs, such as Auger-like scatterings, in which a relaxing electron transfers energy to another electron. This process is promoted to the continuum or electron–hole scatterings, in which the energy of the ES electron transfers to a GS. Quochi et al. observed very fast carrier dynamics at the RT, even at low carrier densities, deducing that the phonon bottleneck is not observable [103]. Gündoğdu et al. reported relaxation times of 450 ± 20 fs for p-doped QDs, 1.4 ± 0.2 ps for n-doped QDs, and 4.8 ± 0.4 ps for undoped QDs, respectively, and the carrier relaxation from the barrier layers to the GS of the QD was strongly enhanced due to the rapid electron-hole scatterings involving the population of the built-in carriers [104].

### 3.2. Device Properties Based on QD Materials

#### 3.2.1. Small Linewidth Enhancement Factor

The chirp is a key parameter for the application of directly modulated lasers, i.e., change of emission wavelength during the direct current modulation. The emission shift originates from a change in the refractive index caused by changing the injection current, resulting in a variation of gain that modifies the phase of the optical mode in the laser cavity. It can be characterized by a linewidth enhancement factor α, defined as $\alpha =-2\frac{\hbar c}{E}\frac{\partial n_{r}/\partial N}{\partial g_{mat}/\partial N}=-2\frac{\hbar c}{E}\frac{{n^{\prime }}_{r}}{g^{\prime }}$, where *c* is the speed of light in a vacuum, *E* is the photon energy, $n_{r}$ is the real part of the complex refractive index, and $g^{\prime }$ is the differential material gain [105]. The α can be calculated by the Kramers–Kronig relation of the gain spectrum. For an ideal QD laser with a perfect Gaussian energy distribution, the differential change of the refractive index is exactly zero at the lasing energy, thus α = 0, i.e., it is chirp-free for the ideal QD laser. In 1999, Newell et al. reported a lower α = 0.1, which is a value that is considerably lower than that of QW lasers [106]. In 2006, Kim et al. theoretically and experimentally studied the effects of the p-type modulation doping on the optical gain, the refractive index change, and the α of the QD lasers. In addition, it was found that p-type doped QD lasers had a better α due to the reduced transparency carrier density [107]. As shown in Figure 13, Duan et al. reported an ultra-low α value of 0.13 that is rather independent of the temperature range (288–308 K) for p-doped QD lasers epitaxially grown on silicon substrates [108]. For undoped QD lasers, the α is increased with the increase in the temperature, which can be attributed to the reduced GS differential gain and the increased refraction index variations. For p-doped QD lasers, the refraction index variation is rather constant with the changing of temperature because of the reduced Auger recombination. The above advances in α makes p-doped QD materials have great potential in low-chirped devices.

#### 3.2.2. Low Threshold Current Density

The threshold current density is an important parameter of a semiconductor laser. Researchers have been working on how to reduce the threshold current density by optimizing the quality of the materials and the device structures. In 1990s, the threshold current density of semiconductor QW lasers was developed to the limit, and the advent of QD lasers makes it possible to further reduce due to the reduced density of states. As shown in Figure 14 [5,19,109,110,111,112,113,114,115,116,117,118,119,120,121,122,123,124,125], in a few years the threshold current density of QD lasers has exceeded the best value for QW lasers. Actually, only two years after the first demonstration of the QD laser, a very low threshold current density of ∼60 A/cm^−2^ was achieved for a QD laser with a vertically coupled QD structure to overcome the GS gain saturation [121]. In 1999, Liu et al. proposed a DWELL structure to obtain nearly 1.3 μm emission, by which a 7 × 10^10^ cm^−2^ high dot density was obtained, and an extremely low threshold current density of 26 A/cm^−2^ was achieved under pulse mode at RT with uncoated cleaved facets [122]. This value is nearly half of the best-reported value of QW lasers. By using the DWELL structure, the low threshold current density of 1.3 μm QD lasers was reported by some research groups [126,127,128,129,130]. With the development of QD technology, the threshold current density of the QD laser further reduced below 10 A/cm^2^, and the lowest internal losses of 0.25 cm^−1^ of CW at RT were reported by Deppe et al. in 2009 [123]. The low threshold current density of QD lasers demonstrates their promise in optical interconnects, not only for minimizing power consumption, but also for avoiding overheating of the lasers and the adjacent electronic circuits.

#### 3.2.3. High Temperature Insensitivity

As early as 1982, when the concept of the QD laser was proposed, Arakawa et al. theoretically predicted that the characteristic temperature of the QD laser was infinite. However, the early reported laser devices exhibit poor temperature stability. Liu et al. showed a T_0_ of 60 K between 10 and 50 °C, but it decreased significantly to 34.5 K between 50 and 80 °C [122]. Huang et al. and Huffaker et al. reported a T_0_ of 41 K near RT and 35 K above 250 K, respectively [20,131]. The temperature stability for the real QD lasers is not significantly improved in comparison to QW lasers near and above RT. One of the main reasons is the carrier thermal excitation out of the GS to the higher QD states or the states in the barriers. So, the energy separation between the GS and the higher ES plays an important role in the operation of the temperature stability for the QD lasers. Chen et al. reported the wide energy separation of 95 meV between the GS and first ES (ES1) by the precise control of growth conditions [132]. The QD lasers with the wider energy separation exhibited high characteristic temperatures of 460 K from 280 to 300 K and 126 K from 300 to 325 K, respectively. Liu et al. reported that another way to improve the temperature stability is to use multilayer QD lasers with the high modal gain [133], in which the T_0_ increasing from 45 to 84 K for N_QD_ is varied from 1 to 3 [76]. A high T_0_ of 110 K in a wide temperature range (15–80 °C) for the QD lasers with no doping incorporation or tunneling injection implementation was demonstrated by Todaro et al. Although the large energy separation between GS and the ES1 of QDs was demonstrated to be beneficial for obtaining a high characteristic temperature of the QD lasers, the large energy separation between radiative transitions is mainly from the electron energy separations. In comparison, the hole-energy separation is much closer, resulting in the injected holes being thermally broadened. The thermal broadening of the holes decreases the GS gain and makes the maximum gain quite temperature sensitive. In order to solve the problem of thermal escape from the GS holes in the QD valence band, in 2002 the research group at the University of Texas introduced p-type modulation doping into the isolation layer of QDs in the active region to increase the occupancy probability of the GS carriers in the valence band, thus improving the T_0_ of the QD lasers [27]. As shown in Figure 15, due to the holes’ easy thermal excitation to a higher level, the number of holes in the valence band is limited for undoped QD lasers. The p-type modulation doping provides a number of additional holes filling into QDs in the valence band level, by which thermal runaway is very difficult for the GS holes. Deppe et al. confirmed that p-doping was very effective in compensating the closely spaced hole levels and had greatly improved the performance of the QD lasers. Subsequently, Shchekin and Deppe obtained a very high T_0_ of 161 K in the temperature range from 0 to 80 °C [134]. After that, Shchekin et al. demonstrated an impressive high T_0_ over 200 K between 0 and 80 °C for a 1.3 μm QD laser, which combined p-type modulation doping with a multilayer QD active region [135]. Invariant output slope efficiency and threshold current in the temperature range of 5–75 °C were obtained for 1.3 µm p-doped self-organized QD lasers by Fathpour et al., as shown in Figure 16a,b [32]. Recently, the infinite characteristic temperature has been also realized in the temperature range from 15 to 50 °C, as shown in Figure 16c,d [136]. A negative characteristic temperature above RT was also reported by Badcock et al. using p-doping and a high-growth-temperature GaAs spacer layer [137,138,139]. The negative T_0_ around RT in p-doped QD lasers is attributed to the change in injection level required to achieve a fixed gain [140,141].

#### 3.2.4. Modulation Characteristics

The operating wavelengths near 1.3 μm of the QD lasers has attracted great interest in relation to the possibility of their use in high-speed optical communications. In order to operate at a higher modulation rate, it is necessary to clarify the origin of the limited modulation bandwidth of the QD lasers and then improve the modulation bandwidth to meet with the network requirements. The small signal modulation of semiconductor lasers is given by following equation:$$M\left(w\right)\propto \frac{w_{r}^{4}}{\left(w^{2}-w_{r}^{2}\right)+w^{2}\gamma^{2}}$$
where $f_{r}=w_{r}/2\pi$ is the relaxation oscillation frequency, and γ is the damping factor given by $\gamma =Kf_{r}^{2}+\tau^{-1}$, with the K factor and the carrier recombination lifetime of τ. The maximum 3 dB bandwidth is defined in terms of the K factor as
$$f_{3dB}=\frac{1}{2\pi }\sqrt{\frac{\left(1-\frac{T}{\tau }\right)+\sqrt{{\left(1-T/\tau \right)}^{2}+T^{2}/\tau^{2}}}{T^{2}}}$$
where $T=K/4\pi^{2}$. In the limit of $\tau^{-1}$→0, the 3 dB bandwidth equation can be reduced to $f_{3dB}=2\sqrt{2}\pi /K$, and the K factor can be represented as $K=4\pi^{2}\left(\tau_{p}+\tau_{cap}\right)$, where $\tau_{p}$ is the cavity photon lifetime and $\tau_{cap}$ is the carrier capture time from the WL to the GS [142,143]. To realize maximum modulation, it is necessary to optimize the photon lifetime. The upper limit modulation bandwidth can be as high as 60 GHz for a single QD-layer laser at 10% QD-size fluctuations and 100% overlap, which does not consider the gain compression with increasing optical power [144]. However, the modulation bandwidth for conventional devices was 6–8 GHz due to the inhomogeneous linewidth broadening and hot-carrier effects, imposing a limit on the performance of the QD lasers [25]. To overcome these problems in conventional QD lasers, tunneling injection and p-doping have been proposed and implemented. The high modulation bandwidth of 25 GHz has been achieved for 1.1 µm p-doped tunnel injection lasers with a T_0_ of 205 K in the temperature range 5–95 °C. For the QD lasers working at the O band, the 1.3 µm wavelength range, Fathpour et al. reported a laser device with a modulation bandwidth of 11 GHz by using a p-doped tunnel injection QD laser structure in 2005 [25], and Tanaka et al. reported a similar modulation bandwidth by using an eight-layer, high-density p-doped QD structure in 2010 [74]. The higher modulation bandwidth has confirmed the potential of QD lasers for uncooled, high-speed fiber-optic communications. Tan et al. reported the first 5 Gb/s data error-free transmission at RT over 4 km of the single-mode fiber and 500 m of the multimode fiber [145]. Two years later, the 16 km transmission was demonstrated at a transfer rate of 2.5 Gb/s at a temperature as high as 85 °C by using a five-layer QD F–P laser structure [146]. A higher modulation bandwidth of 7.4 GHz of 1.3 μm QD lasers has been reported using ten-layer QDs and error-free 8 and 10 Gb/s data transmission at RT [147]. Later, a direct modulation with 5 Gb/s at the temperature range of 15–85 °C and 10 Gb/s at the temperature range of 15–50 °C was demonstrated using a laser with six-layer undoped QDs, which have a saturation modal gain as high as 36.3 cm^−1^ [77]. The modulation bandwidth of p-doped ten-layer QD lasers was obtained with 8 GHz at 20 °C, and it decreased only by 1 GHz at 70 °C. Moreover, a 10 Gb/s direct transfer rate by Otsubo et al. exhibited an extinction ratio of 6.5 dB between 20 and 70 °C without using any current adjustments [148]. Low-driving-current 10 Gb/s operation in the temperature range of 20–90 °C was also achieved by optimizing cavity length and mirror reflectivity [149]. Gerschϋtz et al. first reported the 10 G/s transmission over 21 km in a wide temperature range of 25–70 °C using low threshold current QD distributed feedback lasers [150]. In 2009, Tanaka et al. developed a high-density, five-layer QD laser, which had a density twice that of the conventional QD, and obtained the large net modal gain (7 cm^−1^ per QD layer). The fabricated lasers with F–P ridge waveguides exhibited an extremely temperature-stable operation up to 100 °C and, for the first time, realized 20 Gb/s modulation waveform at RT [151]. The same authors demonstrated a transfer rate of 25 Gb/s using eight-layer high-density QD lasers. These superior characteristics indicate that QD lasers are promising as future light sources in low-cost and low-power-consumption applications [152,153]. Moreover, improving the modal gain of QD lasers is of great significance for achieving a higher speed. So, it is necessary to develop high-density and more-ordered QD arrays, which make it possible to increase the gain and differential gain, and also the use of new schemes for the injection of charge carriers capable of providing more rapid occupation of the QD GS. However, up to now the most commercial optical communication systems rely on InGaAlAs multiple quantum-well lasers, due to the high frequency response and modulation characteristics.

#### 3.2.5. High Optical Feedback Tolerance

The optical isolator is an indispensable part of current optical transmitters in preventing the external reflections back to the laser. To reduce the packaging cost of the lasers, it is advantageous to work without optical isolators [154,155]. The QD lasers have the potential for high external feedback resistance due to lower linewidth enhancement factor, higher damping, and shorter carrier lifetime. In 2003, the behavior of QD lasers under external optical feedback was reported by Su et al. [156]. The critical coherence collapse feedback ratio of the QD DFB laser is −14 dB larger than the QW DFB lasers with typical values between −30 dB and −20 dB. In addition, at a 2.5 Gb/s modulation rate, the signal-to-noise ratio of the QD DFB laser starts to degrade at a feedback ratio of −30 dB, which is about 20 dB higher than a typical QW DFB laser at the same output power and extinction ratio. It is demonstrated that the QD lasers have the potential to realize an isolator-free optical transmitter. Mizutani et al. reported an isolator-free Si-photonics transmitter with a QD laser [157]. A high tolerance to optical feedback of −30 dB against near-end reflections was successfully achieved. Furthermore, the error-free operation of an optical I/O core at 25 Gb/s without optical isolators was demonstrated. In 2020, Huang et al. confirmed that the 1.3 μm QD lasers epitaxially grown on Si exhibit a very high tolerance to optical perturbations, with up to 16% (−8 dB) of the light back reflected to the front facet [158]. The outstanding performance of the QD lasers in the strong feedback situation has demonstrated its possibility of being used in isolator-free 10 Gb/s access networks.

## 4. QD Lasers towards Fiber-Optic Communication

Over the past few decades, tremendous developments have been witnessed in QD lasers: the commercial 10 Gb/s QD lasers can be operated in environments of up to 100 °C. For the realization of low power consumption and a low-cost optical communication light source, a lot of effort and progress has been devoted to fabricating efficient QD lasers. In this section, QD lasers, including F–P lasers, DFB lasers, VCSELs, and QD-SESAM mode-locked lasers are reviewed.

### 4.1. Active Lasers

#### 4.1.1. F–P Lasers

The data communications market requires low-cost, highly-reliable long wavelength F–P lasers. The GaAs-based QD laser is one of the most suitable candidates, due to the high temperature-insensitive properties and the low-cost nature. The threshold current and the efficiency of mass-produced QD F–P lasers produced by QD Laser Inc. have quite weak temperature dependencies in the temperature range of 30 to 100 °C. The optical power of these lasers was more than 10 mW in this temperature range. Long-lifetime reliability is very important for practical applications. A long lifetime of QD lasers provided by QD Laser Company at 85 °C exceeds more than 300,000 h [159]. High-speed lasers are fascinating for their applications towards optic communications and shortening the cavity length, which is an effective way to increase the modulation rate due to the reduced photon lifetime. The shortening of the laser cavity requires higher-gain active materials. As shown in Figure 17, the QDP lasers show superior lasing performance by employing the very short cavity without the facet coating [78]. The P-I characteristics of QDU lasers with different cavity lengths L are shown in Figure 17a,c,e. There are two obvious thresholds observed in Figure 17a. The first at 23 mA (1.31 kA/cm^2^) corresponds to the GS lasing, and the slope efficiency of GS is ~0.24 W/A. With the injection current increased to 50 mA (2.86 kA/cm^2^), the ES lasing can be observed, and the slope efficiency of ES is ~0.41 W/A. The higher slope efficiency of ES is due to the higher gain of the ES band originating from double degeneracy. The inset of Figure 17a shows the lasing profile under various injection currents. Only the GS lasing can be obtained at the low injection current, and the simultaneous GS and ES lasing can be achieved with an injection current increase. On the other hand, only ES lasing is at a further increased current, the reason for which can be attributed to the GS gain saturation and the population increases of ES due to Pauli blocking. As illustrated by Figure 17c,e, only ES lasing can be observed on account of the small GS gain in QDU materials and the shortened cavity length L. In contrast, besides the higher GS gain by the p-doping, the In-Ga intermixing is greatly reduced due to lower Ga vacancies in the p-doped materials, and the high state density of the GS band can be well maintained in the QDP sample. Therefore, the GS lasing in a shorter cavity has been observed in QDP lasers. Figure 17b,d,f show the P−I characteristics of QDP lasers with various L. The GS lasing can be observed with an L of 500 μm, as shown in Figure 17b, and the threshold current was 34 mA (1.94 kA/cm^2^), and the slope efficiency was 0.30 W/A. Compared to undoped lasers, the p-doped lasers exhibited a higher threshold current density and a higher slope efficiency of GS, which can be attributed to the increased carrier–carrier scatterings process and the improved gain characteristics caused by p-doping. As shown in the inset of Figure 17b, only the GS lasing at about 1.3 μm can be found under various injection currents, with the emission intensity over 52 dB, by subtracting the maximum power of the background noise. Compared with the undoped QD lasers, the p-doped laser maintains a good GS lasing with a very short cavity of 400 μm, as shown in Figure 17d, which can be attributed to the reduced gain saturation and the inhibition of the ES lasing by p-doping. It is worth noting that with the injection current further increased, even at >110 mA, the GS emission still maintains a large intensity. We can then conclude that the p-doping is an important technique for the realization of high speed QD lasers. Using a p-doped QD laser structure with a cavity length of 500 μm, Arsenijević et al. reported small-signal bandwidths of 10.51 and 16.25 GHz for the GS and ES, respectively, and large-signal digital modulation capabilities of 15 and 22.5 Gb/s [160].

The global data traffic has rapidly increased in recent years. For the realization of high-speed/throughput optical interconnect with low power consumption in the data-center, the distance between two nodes has become less and less. Optical interconnection on the chip is the ultimate goal to meet these requirements; silicon photonics has been intensively studied in recent years [161,162,163,164]. III-V semiconductors have superior optical properties; so, integration of III-V materials on a Si platform has become one of the most promising techniques [124,165,166,167,168,169]. However, there are a few significant hurdles for epitaxial grown III-V semiconductors on Si substrates: polar III–Vs versus non-polar Si surfaces, the large lattice mismatch and different thermal expansion coefficients, which introduce antiphase boundaries (APBs), high-density threading dislocations (TDs), and thermal cracks [170]. Compared to conventional bulk materials and QW structures, one of the great benefits of QDs is that they are less sensitive to TDs due to the carrier localization [171,172]. The first electrically pumped 1.3 μm InAs/GaAs QD laser epitaxially grown on Si was realized by Wang et al. from UCL [166]. The laser structures were directly grown on Si (001) substrate with 4° off-cut towards the [011] plane. A significant progress towards QD laser directly grown on Si substrate was made by Chen et al. in 2016 [125]. In this work, to prevent the formation of APBs, the InAs/GaAs QD lasers were directly grown on Si (100) wafers with 4° off-cut to the [011] plane. To realize high-quality QD lasers on Si, it is critical to reduce the density of TDs; various growth techniques have been adopted. A 6 nm AlAs nucleation layer was deposited for the initial growth stage, in which three-dimensional growth was suppressed and that provided a superior interface between the Si substrate and the GaAs buffer layer. Then a three-step growth of GaAs buffer layers was introduced, which incorporates 30, 170, and 800 nm thick GaAs grown at 350, 450, and 590 °C, respectively. Most defects were confined in the first 200 nm GaAs buffer region, but high-density TDs (1 × 10^9^ cm^−2^) were still propagated towards the active region. Strained-layer superlattices (SLSs) have been employed as dislocation filter layers to further reduce the density of TDs. Furthermore, a fourfold in situ thermal annealing of the SLSs was implemented, by which the density of TDs reduced to ~10^5^ cm^−2^. In addition, the high-performance electrically pumped CW 1.3 μm InAs/GaAs QD lasers on Si have been achieved with lasing up to 75 °C with a record low threshold current density of 62.5 A/cm^2^ and a high output power exceeding 105 mW at RT. The lifetime over 100,158 h was estimated by continuous-wave operating, and data have been collected for over 3100 h.

#### 4.1.2. Distributed Feedback Lasers

For the long-distance fiber-optic communication, DFB lasers are key devices due to the single-longitudinal-mode oscillation. In 2011, the 1.3 μm QD DFB laser with the wide-temperature-range (−40~80 °C) and a modulation rate of 10.3 Gb/s was reported by Takada et al. using InGaP/GaAs index-coupled gratings [75]. A 10 Gb/s data transmission across 30 km of a single-mode fiber was realized by Stubenrauch et al. in 2014, utilizing an 800 μm long p-doped InGaAs QD DFB laser [173]. Regrowth is detrimental to the formation of high-quality materials for DFB lasers due to the introduction of nonradiation recombination centers. As shown in Figure 18, a 500 μm QD laterally coupled DFB (LC-DFB) structure was fabricated by Li et al. [78]. In this work, the DFB laser has a low threshold current of 30 mA under a CW operation at RT by using p-doped QD structure, and the side-mode suppression ration (SMSR) can be up to 51 dB at a drive current of 55 mA, as shown in the inset of Figure 18c. Due to the wider gain spectrum of QDs, a large tuning range of 140 nm (from 1200 to 1340 nm) was realized for the QD LC-DFB laser, as shown in Figure 18d. As depicted in Figure 19, the wide tunable dual-wavelength lasing operation was realized using QD LC-DFB lasers by delicately defining different periods for the grating structures on the two sides of the laser ridge and combining the reduced laser cavity length [136]. The large tuning range for the two lasing modes can be flexibly tuned in the range of 0.5 to 73.4 nm, corresponding to the frequency difference from 0.10 to 14 THz, which is the largest tuning range by the utilization of single device and hence provides a new opportunity towards the generation of CW THz radiation.

An electrically pumped DFB laser array fabricated in InAs/GaAs QDs grown on Si under RT CW operation was first reported by the researchers from UCL and the Sun Yat-sen University in China [174]. In this work, the InAs/GaAs QDs structure was grown on the n-doped Si (001) substrate with a 4° off-cut towards the [011] plane. As shown in Figure 20, the DFB structure was fabricated by inductively coupled plasma (ICP) during the waveguide etching process, and a λ/4 phase shift in the middle of the etched gratings can be observed. As a result, the first electrically pumped single mode DFB laser array based on InAs/GaAs QD gain material grown on Si has demonstrated lasing at RT under CW pumping. It had a low threshold current of 12 mA and a high SMSR of 50 dB and also exhibited a large wavelength coverage of 100 nm due to the inhomogeneous nature of QDs, as shown in Figure 21.

#### 4.1.3. Vertical Cavity Surface Emitting Lasers

VCSELs can be made in a small footprint, which can operate at very low threshold currents. This test-friendly planar technology has important components for high-data-rate communications. Compared to edge emitting lasers, VCSELs have a circular output-aperture for efficient signal-to-fiber coupling and a more temperature-stable wavelength [175]. The first QD-VCSEL operated at RT was realized in 1996 by Saito et al. [176,177]. The active region of this QD-VCSEL consisted of ten layers of InGaAs QDs with bottom 18-period AlAs/GaAs DBRs and top 14.5-period DBRs. The QD-VCSEL has a CW operating current of 32 mA at RT. The breakthrough of QD-VCSELs came in 1997 [178], when QD-VCSELs with a very low threshold-current (<200 μA) and peak power conversion efficiencies exceeding 10% were fabricated. This was the first experimental QD-VCSEL to rival the best QW-VCSELs. The high efficiency of QD-VCSELs was achieved by placing the QDs in an AlGaAs matrix and increasing the number of dots in each stack. In 2020, Ledentsov et al. reported 850 nm QD VCSELs with a high temperature stability and low threshold currents of 1 mA, operating at the temperature range of 30 to 200 °C. The 25 Gb/s NRZ multi-mode fiber transmission with QD VCSELs was realized at temperatures up to 180 °C [179]. Additionally, for 980 nm QD-VCSELs, the 20 Gb/s NRZ optical eye diagrams at 25 and 85 °C without current adjustment were demonstrated by Hopfer et al. [180], and energy-efficient VCSELs operating at 35 Gb/s at 85 °C with only 139 fJ/bit of dissipated heat were reported by Li et al. [181]. The first GaAs-based 1.3 μm QD-VCSEL was reported by Lott et al. [182], operating at pulsed mode at RT with a threshold current below 2 mA and a differential slope efficiency above 40%. In 2009, the efficient RT CW lasing operation of the 1.3 μm QD-VCSELs for the second-generation optical-fiber communication was achieved by Xu et al. [183]. The threshold current of 6.2 mA (7.9 kA/cm^2^) for the 10 μm diameter QD VCSELs, the differential efficiency of 0.11 W/A, and the maximal output power of 0.85 mW were recorded.

### 4.2. Passive Lasers

#### QD-SESAM Mode-Locked Lasers

Due to their high-power density, high repetition frequency, and narrow pulse duration, the ultrashort pulse lasers can be used in the fields of medical imaging, space ranging, laser weapons, material processing, optical fiber communication, and confidential measurement [184,185,186,187,188]. Ultrashort pulse lasers have mainly been achieved by the active and passive mode-locked methods. In comparison, by using the nonlinear saturated absorber, the passive mode-locking is much easier for obtaining the stable ultrashort pulse lasers with low cost. Recently, QD-SESAMs have emerged as a key candidate for realizing ultrashort pulse lasers due to their board operation bandwidth, fast carrier recovery time, and low saturation fluence [22,189,190]. In 2012, the first 10 GHz repetition rate QD-SESAM mode-locked laser with emission wavelength at 1.55 μm was realized by Zhang et al. using asymmetric In(Ga)As/GaAs DWELL QD structures [41,191]. The TEM image of the QD-SESAM is shown in Figure 22a. Based on this structure, an average output power of 8 mW at a pump power of a 130 mW, 1.4 nm FWHM, and ~2 ps pulse width was realized for the QD-SESAM mode-locked 10 GHz laser, as shown in Figure 22b,c. However, the small modulation depth of 0.4% of these QD-SESAMs limits further improving the mode-locked laser performances. Therefore, optimizing the QD-SESAMs structures to enhance the modulation depth is highly desirable. In 2019, a small saturation intensity of 13.7 MW/cm^2^ and a high nonlinear modulation depth of 1.6% were achieved, simultaneously, by Jiang et al. using a short-period (In_0.20_Ga_0.80_As/In_0.30_Ga_0.70_As)_5_ superlattice (SSL) structure instead of the InGaAs capping layer of QD to achieve a shorter pulse width of an ultrafast laser [192]. The detailed information (the cross-sectional TEM and AFM images) of this QD SSL-SESAM is shown in the inset of Figure 23. The experimental setup of the mode-locked fiber laser with the QDSSL-SESAM is depicted in Figure 23. The cavity length is 24.5 m and consists of a standard single-mode fiber of 23.75 m and a 0.75 m erbium-doped fiber (EDF) as the gain medium. A 980 nm semiconductor DFB laser serves as the pump source, and a 980/1550 nm wavelength division multiplexer (WDM) couples the pump energy to the fiber laser cavity. A polarization insensitive isolator (PI-ISO) and a polarization controller (PC) are used to ensure the unidirectional operation of the fiber laser and optimizing mode-locking state in the cavity, respectively. With the pump power increased to 50 mW, the mode-locking behavior can be observed. In Figure 24a, the output power of this mode-locking laser was linearly increasing with the pump power increased and achieved a slope efficiency of 4.82%. The 3-dB bandwidth of 3.2 nm with a central wavelength of 1556 nm can be observed in the output optical spectra, as shown in the Figure 24b. Figure 24c shows that the signal-to-noise ratio is about 51 dB with a repetition rate of 8.16 MHz, indicating the potential to realize the stable mode-locking operation. Additionally, a pulse duration of 920 fs is achieved, as shown in Figure 24d. The reduced pulse duration of the QD-SESAM, based on SSL capped QDs, can be ascribed to the increased modulation depth and the decreased saturation intensity due to the enhanced carrier relaxation induced by the SSL capping layers. Moreover, all of the mode-locked lasers have long-term stability, by which the high repeatability and the reliability of the SESAMs have been demonstrated.

## 5. Conclusions and Outlook

The III-V QD based lasers have made great progress in many aspects, ranging from materials epitaxy growth and materials characterization to device fabrication and applications. The initial results, including 17 A/cm^2^ ultralow threshold current density, infinite T_0_, high-temperature operation up to 220 °C, and 25 Gb/s direct modulation rate, have unambiguously suggested the greatness of using III-V QDs for the application of fiber-optic communication. Presently, the optimization of III-V QD lasers is still proceeding. For the practical application of fiber-optic communications, challenges still remain but that may in turn bring great opportunities and become the new driving force in these fields. We believe that as the QD lasers could operate without cooling and optical isolators, therefore low-cost, low power consumption, and compact size optical transmitters based on QD lasers are suitable for future optical networks. For low-cost optical components, III-V QD lasers monolithically grown on Si that can be manufactured by complementary metal-oxide-semiconductor (CMOS) compatible processes are an attractive approach. In addition, a new (novel) structure could be introduced/combined into the III-V QD structures, such as the hybrid saturable absorber with van der Waals heterostructures combining graphene or other two-dimensional materials with III-V semiconductors, which offers effective access to the physics, functionalities, and superior performance of optoelectronic devices.

## Figures and Tables

**Figure 1 nanomaterials-12-01058-f001:**
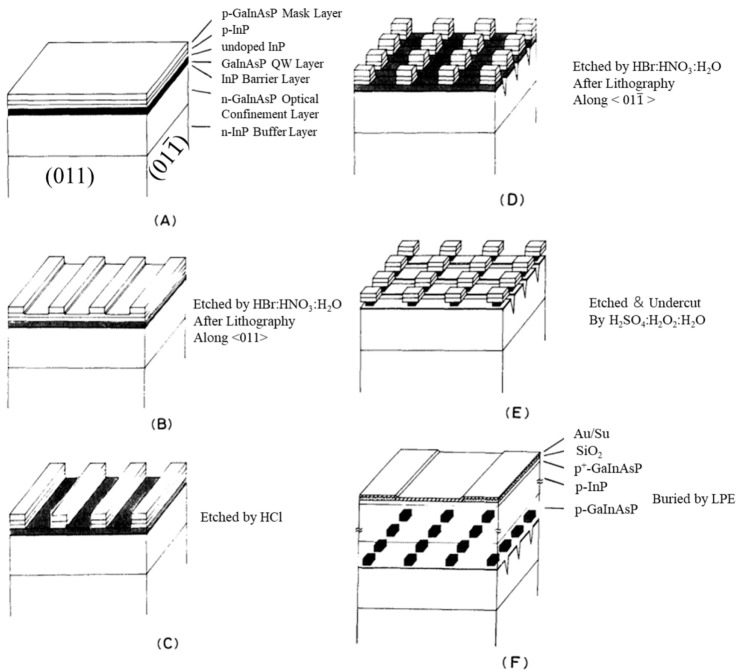
Schematic diagram of the fabrication process for QD structures by using top-down approach. (**A**) GaInAsP/InP QW structures; (**B**) Etched by HBr:HNO_3_:H_2_O along <011> direction; (**C**) Etched by HCl stopped at the surface of the QW layer; (**D**) Etched by HBr:HNO_3_:H_2_O along <01$\bar{1}$> direction; (**E**) QD structures were etched by H_2_SO_4_:H_2_O_2_:H_2_O; (**F**) The QDs was buried by regrowth. Reproduced with permission from [47]. The Physical Society of Japan and The Japan Society of Applied Physics, 1987.

**Figure 2 nanomaterials-12-01058-f002:**
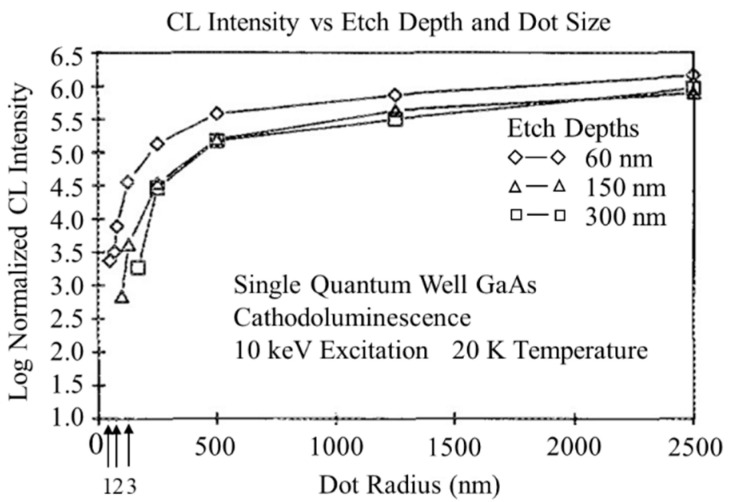
CL intensity of QDs prepared with different etched depths as a function of dot radius. Reproduced with permission from [48]. AIP Publishing, 1989.

**Figure 3 nanomaterials-12-01058-f003:**
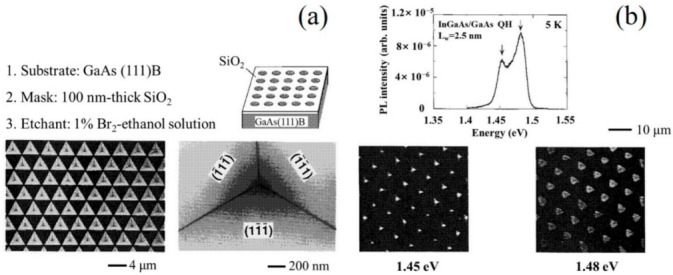
(**a**) SEM images of the patterned substrate and (**b**) PL and CL spectra of QD structures prepared by selectively grown. Reproduced with permission from [49]. AIP Publishing, 1995.

**Figure 4 nanomaterials-12-01058-f004:**
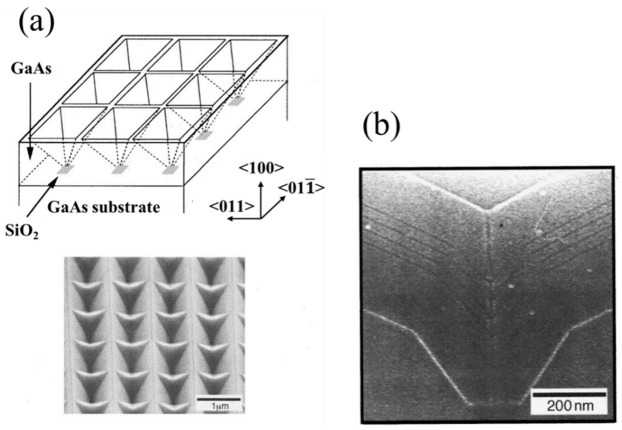
(**a**) Schematic of the patterned GaAs substrate and (**b**) TEM image of four stacks QD structures. Reproduced with permission from [50]. AIP Publishing, 1998.

**Figure 5 nanomaterials-12-01058-f005:**
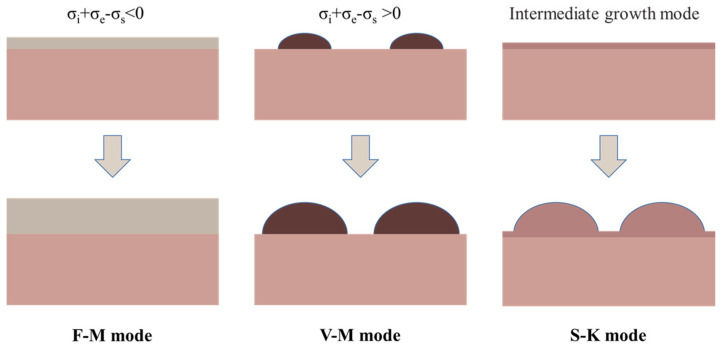
Schematic diagram of the three typical growth modes: F–M mode, V–M mode, and S–K mode.

**Figure 6 nanomaterials-12-01058-f006:**
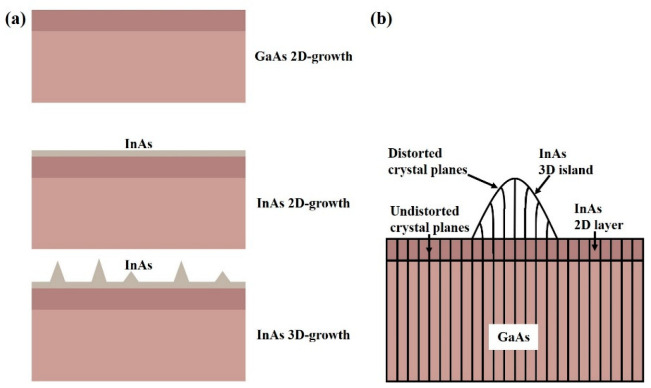
(**a**) Deposition of InAs QDs on the GaAs buffer with S–K growth mode and (**b**) schematic representation of the lattice distortion in a 3D InAs island.

**Figure 7 nanomaterials-12-01058-f007:**
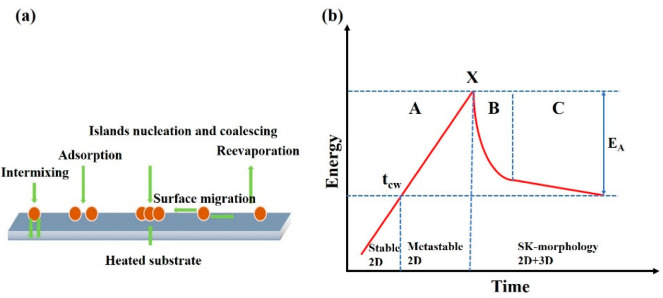
(**a**) Schematics of growth dynamics and (**b**) schematics of total energy versus time for 2D–3D morphology transition.

**Figure 8 nanomaterials-12-01058-f008:**
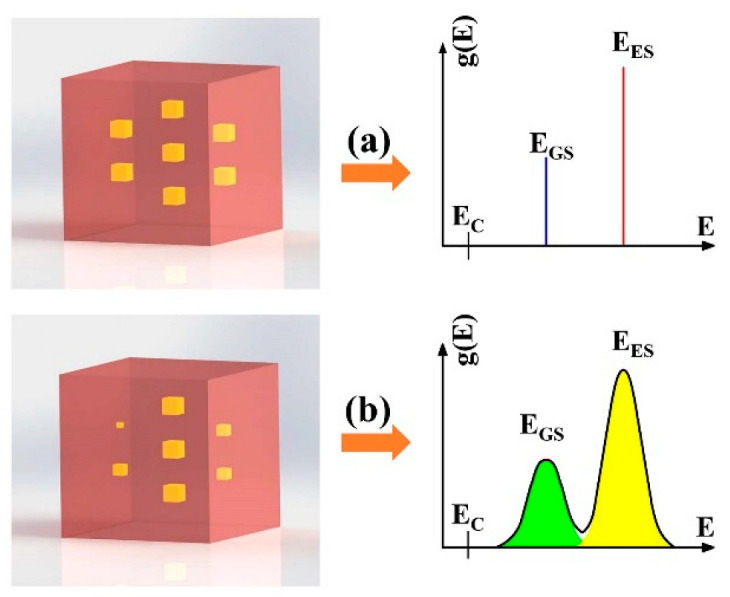
Energy state density of (**a**) ideal and (**b**) real QDs.

**Figure 9 nanomaterials-12-01058-f009:**
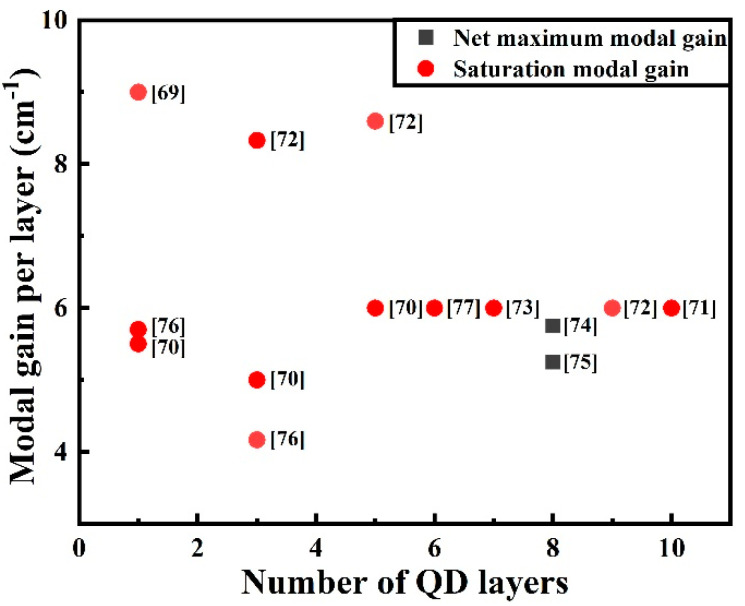
Modal gain of QD lasers calculated per layer in relation to the number of QD layers.

**Figure 10 nanomaterials-12-01058-f010:**
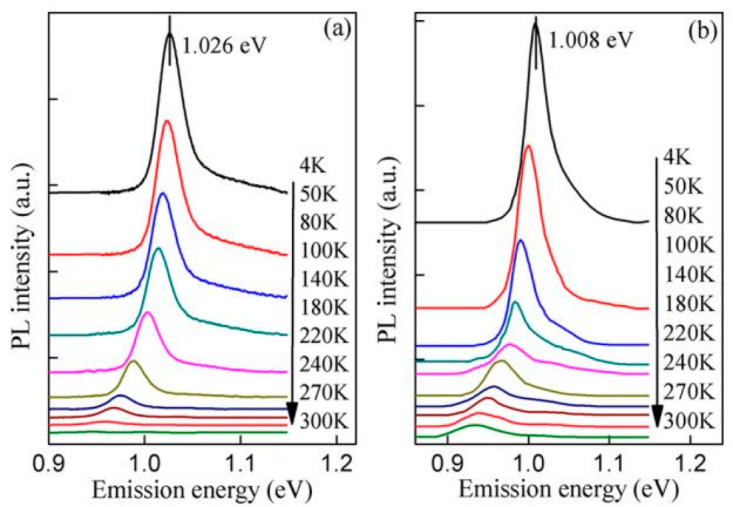
PL spectra measured at 4–300 K from (**a**) the QDU and (**b**) QDP laser structures. Reproduced with permission from [78]. American Chemical Society, 2018.

**Figure 11 nanomaterials-12-01058-f011:**
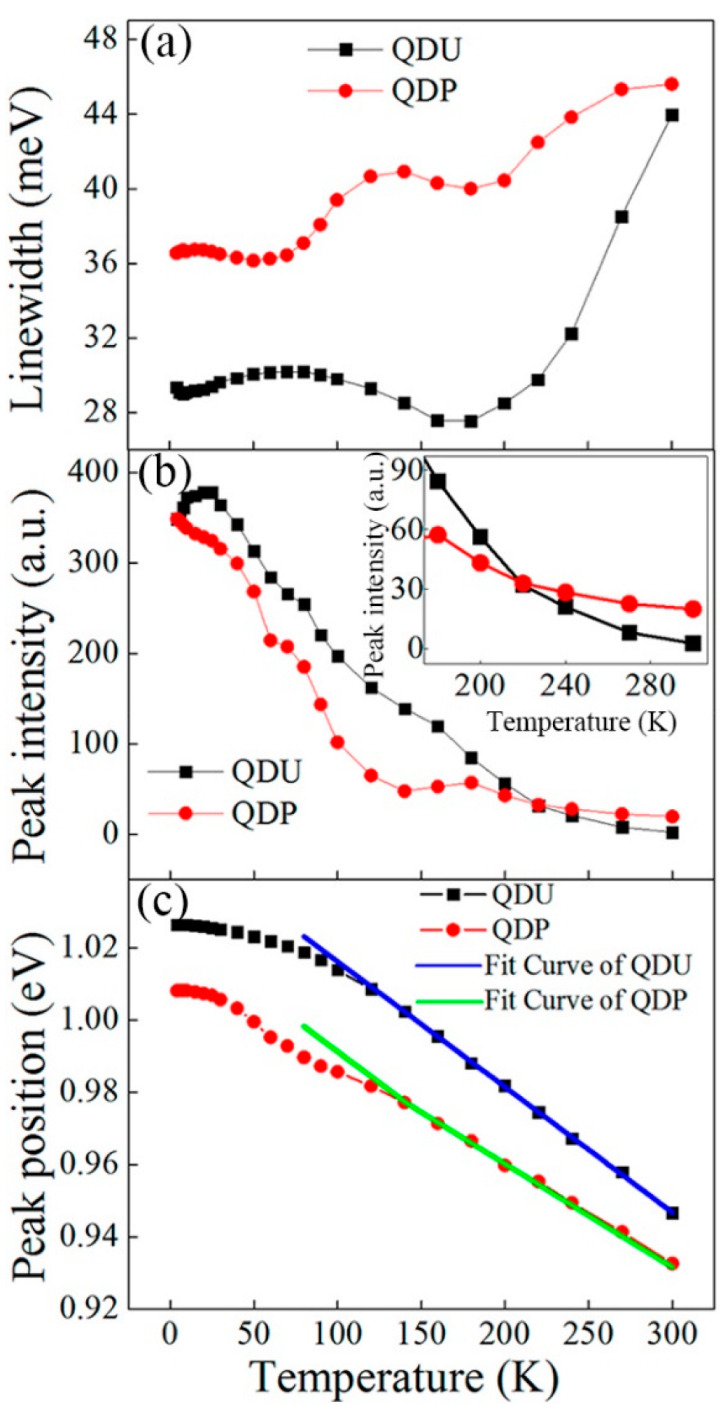
(**a**) PL line width, (**b**) PL peak intensity, and (**c**) PL peak position as functions of temperature. Reproduced with permission from [78]. American Chemical Society, 2018.

**Figure 12 nanomaterials-12-01058-f012:**
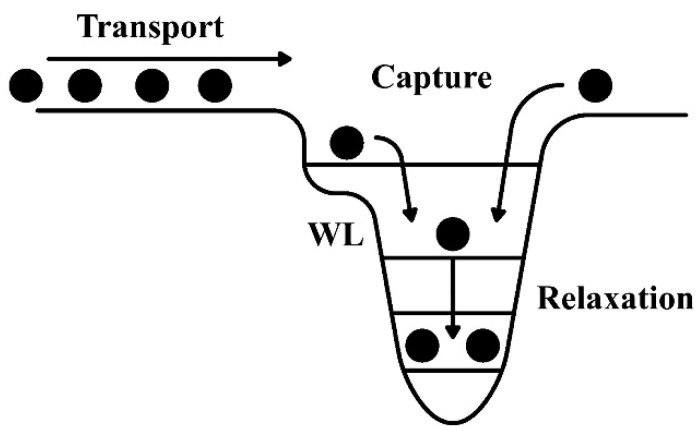
Schematics of carrier dynamics for QDs.

**Figure 13 nanomaterials-12-01058-f013:**
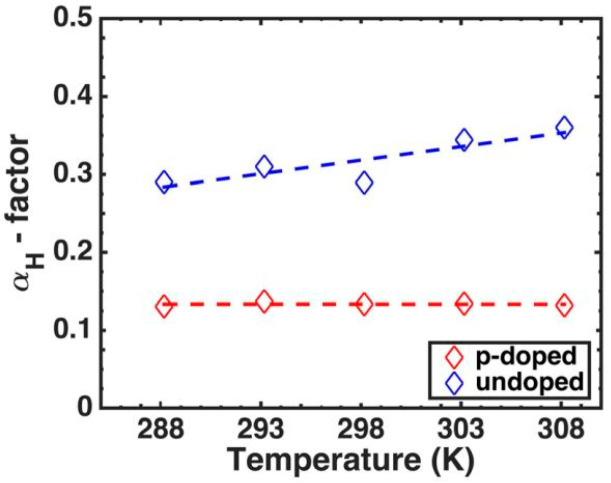
α as a function of temperature for p-doped and undoped QD lasers. Reproduced with permission from [108]. AIP Publishing, 2018.

**Figure 14 nanomaterials-12-01058-f014:**
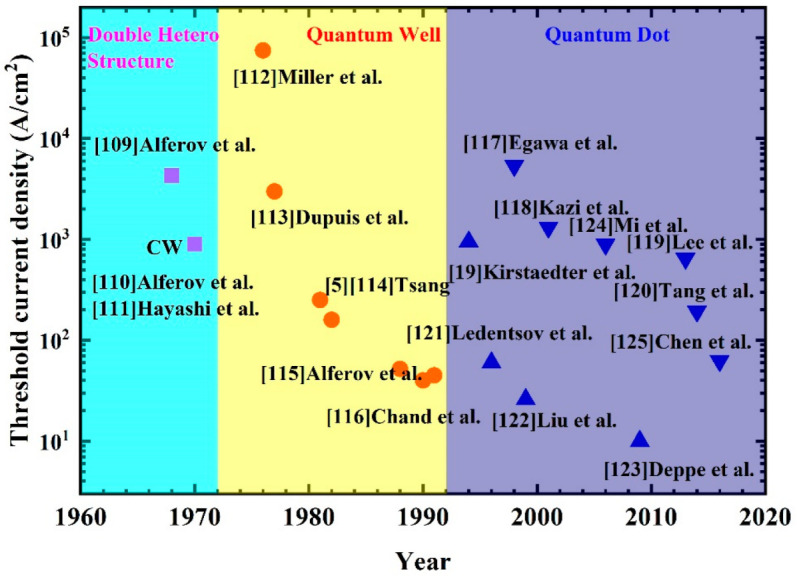
Historical development of heterostructure lasers showing the record threshold current densities at the time of publication (▲QD laser on GaAs; ▼QD laser on Si).

**Figure 15 nanomaterials-12-01058-f015:**
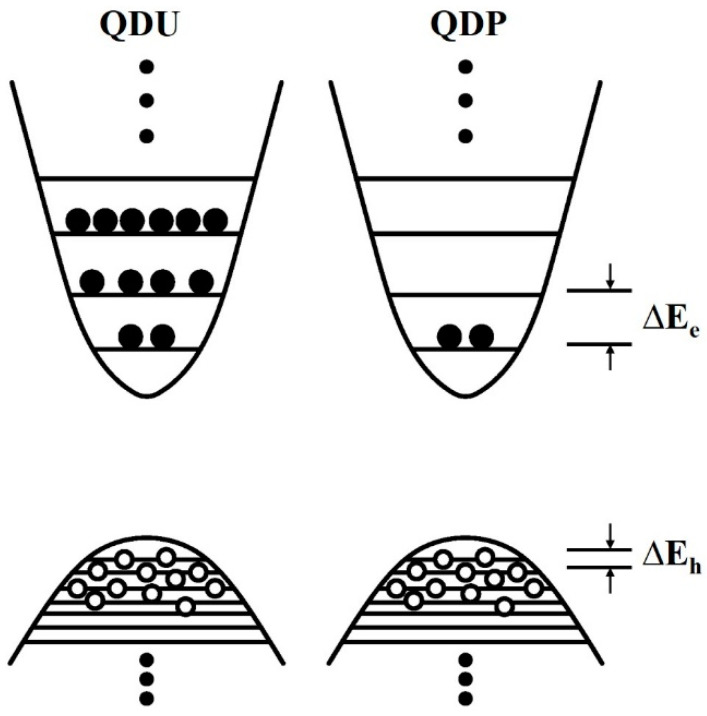
Schematic illustration of the discrete QD energy levels with different built-in carrier distributions.

**Figure 16 nanomaterials-12-01058-f016:**
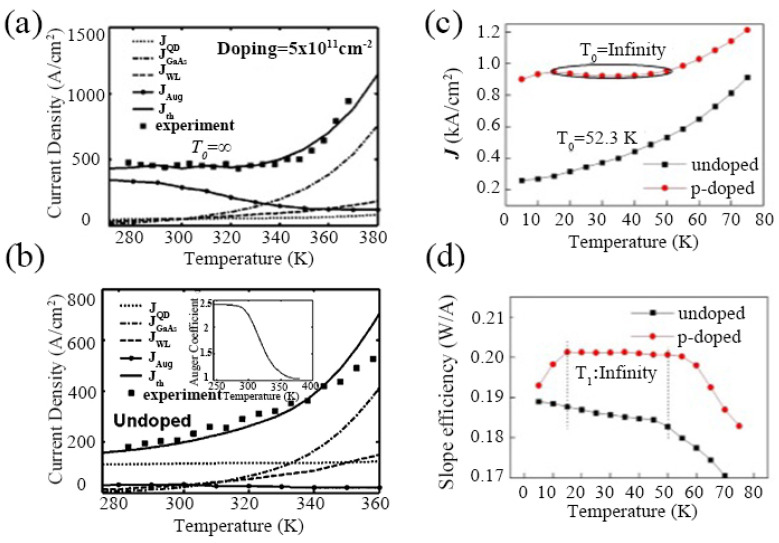
Variation of calculated and measured threshold current density in (**a**) QDP and (**b**) QDU lasers, respectively. Reproduced with permission from [32]. AIP Publishing, 2004. (**c**) Temperature dependences of threshold current density for QDU and QDP lasers. (**d**) Slope efficiency for QDU and QDP lasers. Reproduced with permission from [136]. Springer Nature, 2018.

**Figure 17 nanomaterials-12-01058-f017:**
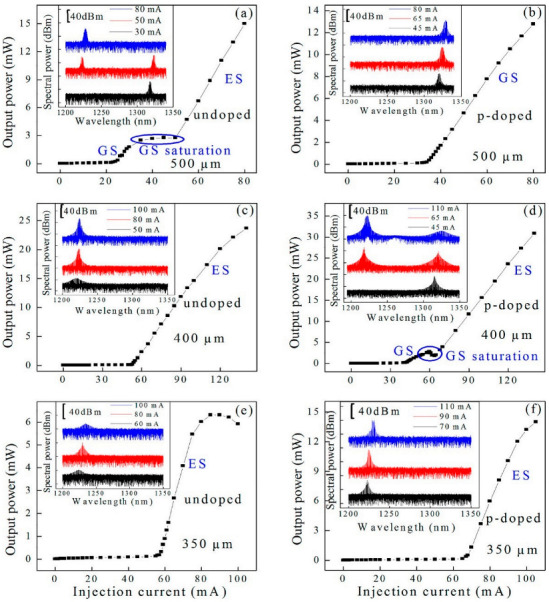
P-I characteristics with different L for QDU and QDP lasers. (**a**,**c**,**e**) for QDU lasers with 500, 400, and 350 μm; (**b**,**d**,**f**) for QDP lasers with 500, 400, and 350 μm, respectively. Inset: Corresponding lasing spectra of the device under various injection currents. Reproduced with permission from [78]. American Chemical Society, 2018.

**Figure 18 nanomaterials-12-01058-f018:**
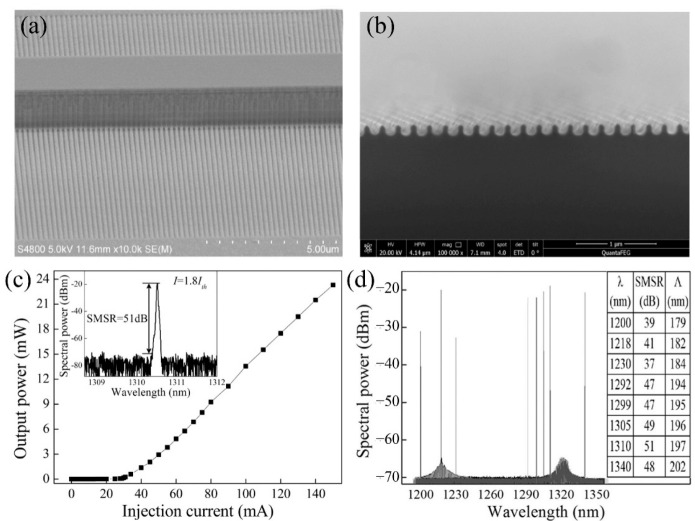
(**a**,**b**) SEM images of the LC-DFB laser with the first-order grating. (**c**) P-I characteristic of a p-doped QD DFB laser under CW operation at RT. Inset: Lasing spectrum of a p-doped QD DFB laser measured at 1.8Ith, and (**d**) lasing spectra of p-doped QD DFB lasers with different grating periods. Inset: Detailed information on the LC-DFB lasers, including the lasing wavelengths, SMSR, and Bragg periods. Reproduced with permission from [78]. American Chemical Society, 2018.

**Figure 19 nanomaterials-12-01058-f019:**
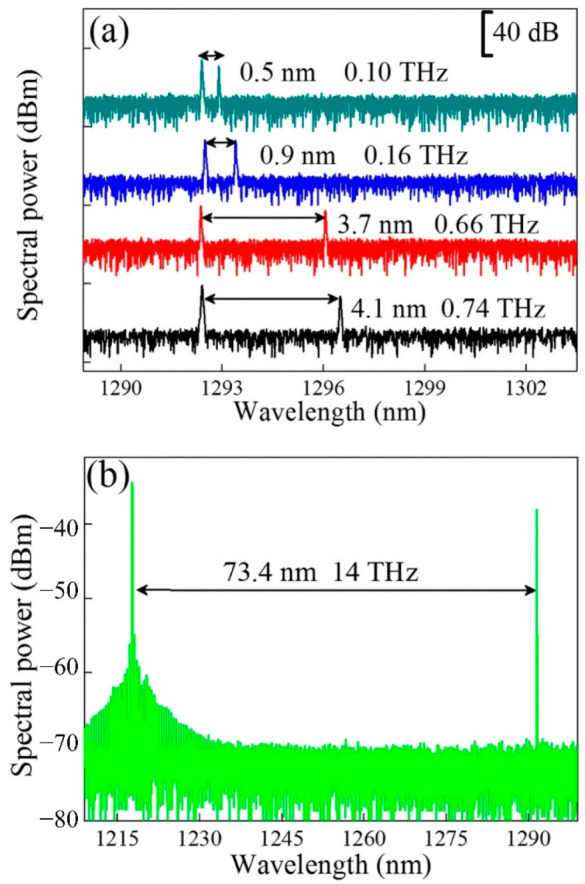
Output spectra of the dual-mode LC-DFB lasers. (**a**) Emission spectra of the dual-wavelength LC-DFB laser with a different grating period. (**b**) Wide spacing of dual-mode lasing spectra of the LC-DFB laser. Reproduced with permission from [136]. Springer Nature, 2018.

**Figure 20 nanomaterials-12-01058-f020:**
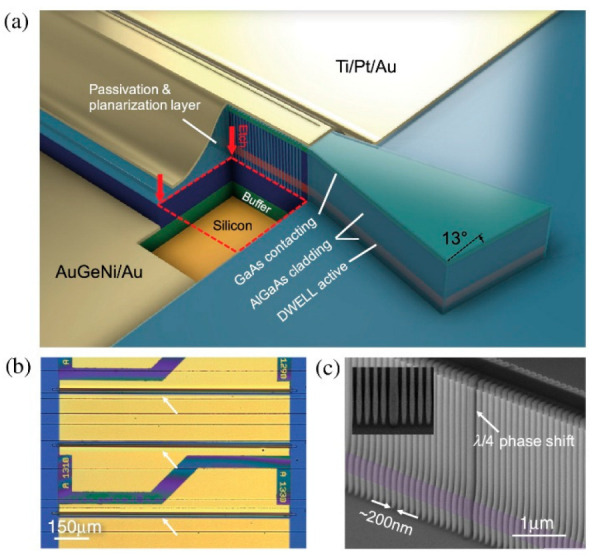
DFB laser array on silicon. (**a**) Cutaway schematic of the DFB lasers (not to scale). (**b**) Regional microscope image of the DFB laser array on silicon. (**c**) High-resolution SEM image of the gratings with a λ/4 phase shift in the middle from a test run. Reproduced with permission from [174]. The Optical Society, 2018.

**Figure 21 nanomaterials-12-01058-f021:**
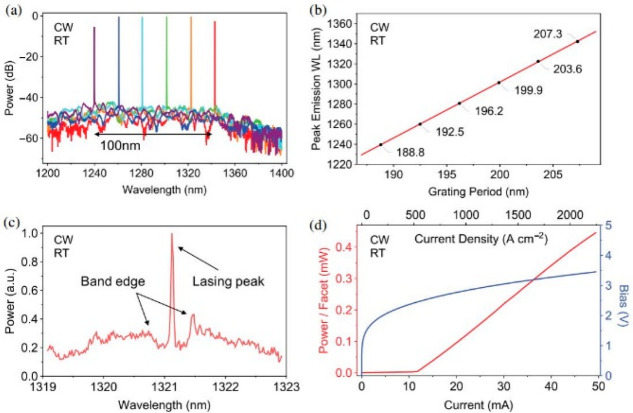
CW test results of the silicon-based DFB laser array at RT. (**a**) Optica spectra of a DFB laser array with different grating periods. (**b**) Peak emission wavelengths of the DFB array plotted against grating period values. (**c**) Zoomed-in optical spectrum of a single DFB laser operating just below threshold. (**d**) Light-current-voltage curve of a single 1 mm long silicon-based DFB laser. Reproduced with permission from [174]. The Optical Society, 2018.

**Figure 22 nanomaterials-12-01058-f022:**
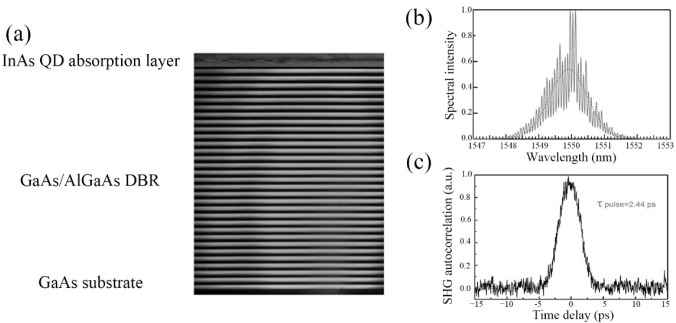
(**a**) TEM image of the 1.55 μm QD-SESAM. (**b**) Laser output of optical power spectrum and (**c**) second harmonic generation autocorrelation trace. Reproduced with permission from [41]. Springer Nature, 2012.

**Figure 23 nanomaterials-12-01058-f023:**
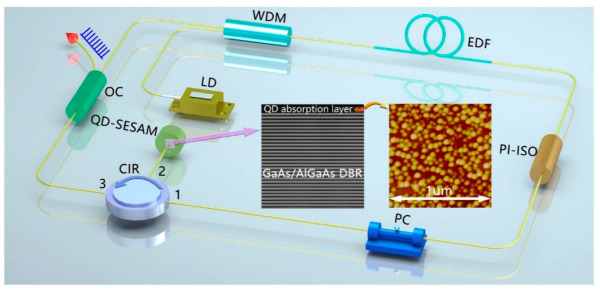
Experimental setup of mode-locked fiber laser with 1550 nm SESAM. Inset: TEM image of the QD-SESAM and AFM image of the 1550 nm QDs. Reproduced with permission from [192]. Springer Nature, 2019.

**Figure 24 nanomaterials-12-01058-f024:**
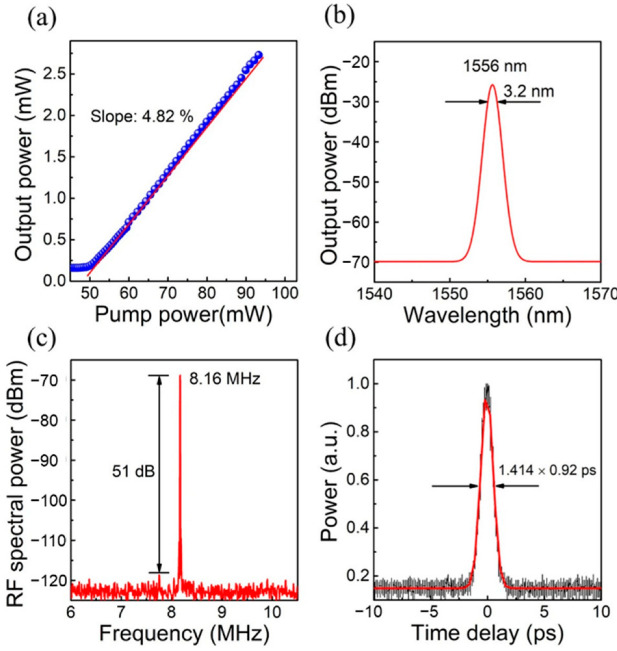
Characteristics of the mode-locked developed fiber laser. (**a**) Output power versus pump power. (**b**) Output optical spectra. (**c**) RF spectrum of the mode-locked fiber laser. (**d**) Autocorrelation trace. Reproduced with permission from [192]. Springer Nature, 2019.

## Data Availability

Not applicable.

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
