# Peer review of "Recent Developments of Quantum Dot Materials for High Speed and Ultrafast Lasers"

_nanomaterials, 2022, doi:10.3390/nano12071058_

Round 1

Reviewer 1 Report

The manuscript "Recent development of Molecular Beam Epitaxy growth of high speed and ultrafast quantum dot lasers towards fiber-optic
communication" by Yao et al is a review concerning the quantum dot laser suitable for fiber-optic communication.

The manuscript is well organized and the sections well written. In my opinion just few point should be addressed:

1) line 104, abbreviations F-P, DFB, VCSEL were not introduced before

2) quality of few figures is poor (e.g. figures 5 and 6). If possible please improve it.

The authors can also consider to add in section 2.2.2 few words about Droplet epitaxy deposition technique (see e.g. Journal of Vacuum Science & Technology B 11, 787 - 1993 - doi:10.1116/1.586789), that allows for the fabrication of strain-free quantum dot laser (e.g. Applied Physics Letters 93, 203110 - 2008, DOI:10.1063/1.3026174). This technique can be used also for the fabrication of InAs QDs with precise control of density and size (e.g.  Appl. Phys. Lett. 118, 133102 -2021- doi: 10.1063/5.0045776)

Author Response

Response to the Reviewers’ comments:

Reviewer 1:

The manuscript "Recent development of Molecular Beam Epitaxy growth of high speed and ultrafast quantum dot lasers towards fiber-optic communication" by Yao et al is a review concerning the quantum dot laser suitable for fiber-optic communication.

The manuscript is well organized and the sections well written. In my opinion just few point should be addressed:

Response: Lots of thanks for the affirmation on our work. We thank the reviewer for reading, commenting, and evaluating our manuscript. The point-by-point answers to the questions can be found below.

Comment 1: line 104, abbreviations F-P, DFB, VCSEL were not introduced before.

Response: Thanks for the reviewer’s suggestion. An introduction to the abbreviations for F-P, DFB, VCSELs were added to the revised manuscript. We have added “Fabery-Perot (F-P) lasers, Distributed Feedback (DFB) lasers, Vertical-Cavity Surface-emitting lasers (VCSELs),” in the revised paper. (Page 3, started from line 108). In addition, we have carefully checked through the manuscript and corrected all similar small problems.

Comment 2: quality of few figures is poor (e.g. figures 5 and 6). If possible please improve it.

Response: We greatly appreciate the reviewer’s suggestion. The high quality of the pictures has been updated in the revised manuscript. (Page 7, figures 5 and 6)

Comment 3: The authors can also consider to add in section 2.2.2 few words about Droplet epitaxy deposition technique (see e.g. Journal of Vacuum Science & Technology B 11, 787 - 1993 - doi:10.1116/1.586789), that allows for the fabrication of strain-free quantum dot laser (e.g. Applied Physics Letters 93, 203110 - 2008, DOI:10.1063/1.3026174). This technique can be used also for the fabrication of InAs QDs with precise control of density and size (e.g.  Appl. Phys. Lett. 118, 133102 -2021- doi: 10.1063/5.0045776)

Response: Thanks for the reviewer’s comments. As the reviewer’s suggestions, the literature of “Koguchi N, et al. New selective molecular-beam epitaxial growth method for direct formation of GaAs quantum dots. Journal of Vacuum Science & Technology B, 1993, 11, 787”; “Mano T, et al. GaAs/AlGaAs quantum dot laser fabricated on GaAs (311) A substrate by droplet epitaxy. Applied Physics Letters, 2008, 93, 203110”; “Tuktamyshev A, et al. Telecom-wavelength InAs QDs with low fine structure splitting grown by droplet epitaxy on GaAs(111)A vicinal substrates. Applied Physics Letters, 2021, 118, 133102” have been cited as reference 55, 56, and 57 in the revised manuscript, respectively. These articles were added in section 2.2.2 could improve the preparation about self-assembled QD technology. Also, we have added “In addition, It is worth mentioning that the strain-free GaAs/AlGaAs QD can be fabricated by the droplet epitaxy technique, which has a definite advantage enablinga large number of QD layers with keeping highquality since there is no strain induced dislocation. The QD density and size can also be precise control by the droplet epitaxy technique on vicinal GaAs(111)A substrates, which the fine structure splitting as low as 16 μeV was realized, thus making them suitable as photon sources in quantum communication networks using entangled photons.” (Page 6, started from line 234)

Reviewer 2 Report

The authors give an overview of the development in III-V QD lasers at 1.3 μm for fiber-optic communication. They cover topics about the QD growth and the laser properties, as well as the device performance. Concerning the MBE growth of QDs, they make a brief introduction in section 2.2.2, without any reference to recent developments (only one cited work from the last 5 years) as the title of the manuscript misleadingly implies. Concerning the laser properties, the following topics are discussed: the modal gain for different structure designs and optical confinement factors, the temperature dependence of PL for undoped and p-doped QDs (effect of p-doping on carrier recombination), the lower linewidth enhancement factor in p-doped QD lasers owing to a reduced transparency carrier density, the lower threshold current in QD lasers, the p-doping for the compensation of escaped holes from the QDs, the development of modulation bandwidth (the references here are at least ten year old; could the authors add more recent ones?) and, finally, the high tolerance to optical feedback for isolation-free QD lasers. The manuscript closes with an overview of the performance of various laser types: FP lasers (comparison of undoped and p-doped structures; monolithic integration of III-V QDs on Si), DFB lasers (LC-DFB, and DFB on Si), VCSELs, and QD-SESAM lasers.

I believe that this manuscript makes a good complement to other recent review papers on QD lasers (e.g. see J. C. Norman et al., IEEE Journal of Quantum Electronics, vol. 55, no. 2, pp. 1-11, April 2019, Art no. 2000511, and Shujie Pan et al 2019 J. Semicond. 40 101302) and should be accepted for publication. However, the title is misleading because the manuscript does not contain significant information about recent developments in the MBE growth of QD lasers (with the exception of Chen et al. [147] about F-P lasers). Thus, I strongly suggest that the title should be appropriately modified to accurately reflect the contents of the manuscript.

Author Response

Reviewer 2:

Comment 1: The authors give an overview of the development in III-V QD lasers at 1.3 μm for fiber-optic communication. They cover topics about the QD growth and the laser properties, as well as the device performance. Concerning the MBE growth of QDs, they make a brief introduction in section 2.2.2, without any reference to recent developments (only one cited work from the last 5 years) as the title of the manuscript misleadingly implies. Concerning the laser properties, the following topics are discussed: the modal gain for different structure designs and optical confinement factors, the temperature dependence of PL for undoped and p-doped QDs (effect of p-doping on carrier recombination), the lower linewidth enhancement factor in p-doped QD lasers owing to a reduced transparency carrier density, the lower threshold current in QD lasers, the p-doping for the compensation of escaped holes from the QDs, the development of modulation bandwidth (the references here are at least ten year old; could the authors add more recent ones?) and, finally, the high tolerance to optical feedback for isolation-free QD lasers. The manuscript closes with an overview of the performance of various laser types: FP lasers (comparison of undoped and p-doped structures; monolithic integration of III-V QDs on Si), DFB lasers (LC-DFB, and DFB on Si), VCSELs, and QD-SESAM lasers.

Response: We appreciate the reviewer's affirmation on our work indeed. We thank the reviewer for reading, commenting, and evaluating our manuscript. As the reviewer’s suggestions, the literature of “J. Kwoen, et al., Opt. Express, 2019, 27(3):2681-2688.”, “J. Yang, et al., J. Phys. D: Appl. Phys., 2021, 54:035103.”, “C. R. Fitch, et al., IEEE J. Sel. Top. Quantum Electron., 2022, 28(1):1900210” on the recent developments of QD lasers have been cited as reference 57, 58, and 59 in the section 2.2.2 of revised manuscript, respectively. (Page 6, line 229). In addition, more literatures of “A. E. Zhukov, et al., Tech. Phys. Lett., 2019, 45(8):847-849.”, “Y. Wan, et al., Laser Photonics Rev., 2021, 15:2100057” have been cited as reference 150, and 151 in the section 3.2.4. (Page 17, line 580) in the revised paper.

Comment 2: I believe that this manuscript makes a good complement to other recent review papers on QD lasers (e.g. see J. C. Norman et al., IEEE Journal of Quantum Electronics, vol. 55, no. 2, pp. 1-11, April 2019, Art no. 2000511, and Shujie Pan et al 2019 J. Semicond. 40 101302) and should be accepted for publication. However, the title is misleading because the manuscript does not contain significant information about recent developments in the MBE growth of QD lasers (with the exception of Chen et al. [147] about F-P lasers). Thus, I strongly suggest that the title should be appropriately modified to accurately reflect the contents of the manuscript.

Response: Lots of thanks for the affirmation on our work. Thanks for the reviewer’s suggestion. The title of “Recent development of Molecular Beam Epitaxy growth of high speed and ultrafast quantum dot lasers towards fiber-optic communication” has been replaced by “Recent developments of quantum dot materials for high speed and ultrafast lasers” in the revised paper. And the literature of “J. C. Norman, et al., IEEE J. Quantum Electron., 2019, 55(2):2000511.”, “V. Cao, et al., Front. Phys., 2022, 10:839953.” have been cited as reference 45 and 46 in the introduction. (Page 3, line 101)

Reviewer 3 Report

Dear authors,

congratulations to this nice paper.
As this represents a comprehensive overview paper some topics still have to be addressed:

MBE growth:
The authors should compare the QD MBE results with results obtained using MOVPE systems and explain the advantages using MBE systems for QD growth.

1550nm wavelength range:
The authors should also include 1550nm QD results achieved with InP substrates.

modulation characteristics:
It should at least be mentioned that 1300nm QW based devices on InP substrates so far clearly outperform respective QD based devices on GaAs regarding maximum frequency response and modulation characteristics. Therefore most commercial optical systems rely on InGaAlAs MQW devices.

Best regards

Author Response

Reviewer 2:

congratulations to this nice paper.
As this represents a comprehensive overview paper some topics still have to be addressed:

Response:. We appreciate the reviewer's affirmation on our work indeed. We have made the detailed point-by-point answers to all the comments as follows

Comment 1: MBE growth: The authors should compare the QD MBE results with results obtained using MOVPE systems and explain the advantages using MBE systems for QD growth.

Response: Thank you for your instructive suggestions. The description of the comparison of MBE and MOVPE techniques for QDs growth was added in section 2 line 229, We have added “MBE and Metal-Organic Vapor Phase Epitaxy (MOVPE) are the two sophisticated techniques widely used in compound semiconductor thin film growth. For the self-assembled QD under S-K growth mode, most high-quality QD structures are based on MBE technology due to epitaxy structure with composition and doping profiles well controlled at nanometer scale, while MOVPE is suited for the preparation of mass production structures” in the revised manuscript. (Page 6, started from line 229)

Comment 2: 1550nm wavelength range: The authors should also include 1550nm QD results achieved with InP substrates.

Response: Thanks for the reviewer’s suggestion. The description of the InP based 1550 nm QD results was added in line 96, “An InP substrate has the advantage to realize longer wavelength InAs QDs due to the smaller lattice mismatch (3%), instead of GaAs substrates. Therefore, 1.55 μm InP-based quantum dash lasers were reported by many research groups [42-44]” in the revised manuscript. (Page 2, started from line 96)

References added:

[42] Saito H, et al. Ground-state lasing at room temperature in long-wavelength InAs

quantum-dot lasers on InP(311)B substrates. Applied Physics Letters, 2001, 78, 267.

[43] Schwertberger R, et al. Long-Wavelength InP-Based Quantum-Dash Lasers. IEEE Photonics Technology Letters, 2002, 14, 735.

[44] Ukhanov A A, et al. Orientation dependence of the optical properties in InAs quantum-dash lasers on InP. Applied Physics Letters, 2002, 81, 981.

Comment 3: modulation characteristics: It should at least be mentioned that 1300nm QW based devices on InP substrates so far clearly outperform respective QD based devices on GaAs regarding maximum frequency response and modulation characteristics. Therefore, most commercial optical systems rely on InGaAlAs MQW devices.

Response: Thanks for the reminding of the reviewer. The description about the modulation characteristics of the InP-based QW was added in the revised manuscript, “However, up to now the most commercial optical communication systems rely on InGaAlAs multiple quantum well lasers, due to the high frequency response and modulation characteristics.” (Page 17, started from line 584)

Reviewer 4 Report

Reviewers comment manuscript MDPI nanomaterials-1620377

Title: Recent development of Molecular Beam Epitaxy growth of high speed and ultrafast quantum dot lasers towards fiber-optic communication

Authors: Zhonghui Ya, Cheng Jiang, Xu Wang, Hongmei Chen, Hongpei Wang, Liang Qin and Ziyang Zhang

The paper is a review article that presents a research overview of quantum dot (QD) materials and lasers. The article describes  material growth methods, properties of grown QD materials and performance of QD lasers.

The paper gives a helpful overview of the potential benefits of QD materials and the recent advances in the field. I recommend it for publication after some reorganization of the paper and minor corrections to the language.

I suggest the title is shortened to e.g. “Recent developments of quantum dot materials for uncooled high speed lasers”. The paper does not focus on recent development of Molecular Beam Epitaxy (MBE). In fact MBE is only mentioned in the title and the abstract whereas e.g. references 44-48 are using MOCVD. Instead it is better that the title reflects the potential benefits of QD materials for uncooled operation.

In the introduction, I recommend the authors give a motivation for the review, citing previous review articles. I suggest that the historical overview (lines 52-95), currently in the introduction, is merged into the other sections, e.g. 3.2.3, 3.2.4 and 4.1.1.

There are several grammatical and spelling errors that need to be corrected before publication e.g., on lines 222, 382, 398, 405, 410, 426, 478, 479, 772, 783, 829.

Author Response

Reviewer 4:

Title: Recent development of Molecular Beam Epitaxy growth of high speed and ultrafast quantum dot lasers towards fiber-optic communication

Authors: Zhonghui Yao, Cheng Jiang, Xu Wang, Hongmei Chen, Hongpei Wang, Liang Qin and Ziyang Zhang

The paper is a review article that presents a research overview of quantum dot (QD) materials and lasers. The article describes  material growth methods, properties of grown QD materials and performance of QD lasers.

The paper gives a helpful overview of the potential benefits of QD materials and the recent advances in the field. I recommend it for publication after some reorganization of the paper and minor corrections to the language.

Response: Lots of thanks for the affirmation on our work. We thank the reviewer for reading, commenting, and evaluating our manuscript. The point-by-point answers to the questions can be found below.

Comment 1: I suggest the title is shortened to e.g. “Recent developments of quantum dot materials for uncooled high speed lasers”. The paper does not focus on recent development of Molecular Beam Epitaxy (MBE). In fact MBE is only mentioned in the title and the abstract whereas e.g. references 44-48 are using MOCVD. Instead it is better that the title reflects the potential benefits of QD materials for uncooled operation.

Response: Thank you for your instructive suggestions. The title of “Recent development of Molecular Beam Epitaxy growth of high speed and ultrafast quantum dot lasers towards fiber-optic communication” has been replaced by “Recent developments of quantum dot materials for high speed and ultrafast lasers” in the revised paper.

Comment 2: In the introduction, I recommend the authors give a motivation for the review, citing previous review articles. I suggest that the historical overview (lines 52-95), currently in the introduction, is merged into the other sections, e.g. 3.2.3, 3.2.4 and 4.1.1.

Response: Thanks for the reviewer’s suggestion. The literature of “J. C. Norman, et al., IEEE J. Quantum Electron., 2019, 55(2):2000511.”, “V. Cao, et al., Front. Phys., 2022, 10:839953.” have been cited as reference 45 and 46 in the introduction. (Page 3, line 101)

Comment 2: There are several grammatical and spelling errors that need to be corrected before publication e.g., on lines 222, 382, 398, 405, 410, 426, 478, 479, 772, 783, 829.

Response: We greatly appreciate the reviewer’s suggestion. We have corrected the grammatical and spelling errors in the revised manuscript, and we have carefully checked and improved the English writing in the revised manuscript.

Reviewer 5 Report

The title of the presented review states the development of MBE for the creation of laser heterostructures. However, this thesis was not disclosed in the process of preparing this contribution. The Abstract section contains phrase "development of molecular beam epitaxial (MBE) growth methods", while Section 2 in the text of the paper compares different technological approaches getting QD-structures which can be use either MOCVD and MBE methods, as well as LPE too.

The Conclusion section does not summarise experimental results discussed in the text that does not correspond to the declared title.

This review contained a set of slip of pen, some figures does not contain sufficient decriptions, cited materials does not support by number of Reference.

This paper needs major revision. 

Author Response

Reviewer 3:

The title of the presented review states the development of MBE for the creation of laser heterostructures. However, this thesis was not disclosed in the process of preparing this contribution. The Abstract section contains phrase "development of molecular beam epitaxial (MBE) growth methods", while Section 2 in the text of the paper compares different technological approaches getting QD-structures which can be use either MOCVD and MBE methods, as well as LPE too.

The conclusion section does not summarise experimental results discussed in the text that does not correspond to the declared title.

This review contained a set of slip of pen, some figures does not contain sufficient decriptions, cited materials does not support by number of Reference.

This paper needs major revision.

Response: Thanks for the reviewer's comments, but we haven't compared the MBE technology with MOCVD (MOVPE) or LPE technologies on the QD materials and devices through the entire paper. If we have done any of that, the other reviewers couldn't suggest us to add some descriptions on the comparison between QD MBE results and QD MOVPE results. In addition, for a review article, if the figures are reprinted from those papers already published, we need to get the reprint permission from the publishers and also need to announce each corresponding reference at the end of each figure caption, but if the figures haven't published, we don't need to do that.

Reviewer 6 Report

Review written by Yao et al. is devoted to one of the most actively developing areas of semiconductor physics and technology on the development of ultrafast lasers with quantum dots grown by molecular beam epitaxy. Such lasers are essential for the development of fiber-optic communication. Therefore, the relevance of the review is beyond doubt.

The review is based on the classical scheme, and in its first part, the features of various technologies for the formation of quantum dots (primarily InAs) are considered. The authors then describe in detail the fundamentals of the physics of semiconductor quantum dot (QD) lasers, including various phenomena in nanoheterostructures with strong carrier localization that determine a high performance of the QD laser. It should be noted that this chapter is written at a much higher level than the first technological chapter, which is more like an entry-level student essay. Finally, in the final chapter, the authors provide an excellent overview of the state-of-the-art applications of various QD lasers in fiber optic communication devices. In my opinion, this is the most successful and valuable chapter of the review.

To write the review, the authors used a wide range of sources, including both initial and most recent work on quantum dot lasers. I am sure that this review will be interesting and useful to a wide range of readers of the journal, including highly qualified specialists and students.

Most of my comments are technical and are marked in the pdf as comments and suggested changes. The most significant remark concerns the description in the technological chapter (on page 5) of various growth mechanisms:

I would recommend adding the elastic energy term to the total surface energy (the sum of all energies in the film), which determines the observed epitaxial growth mechanism. This energy depends on the layer thickness and, therefore, determines the transition from a 2D- to 3D growth mechanism in the Stranski-Krastanow mode. In addition, it would be desirable to add the energetic inequalities to Figure 5 and illustrate the change of elastic energy in Figure 7. In my opinion, this addition will raise the level of analysis of surface processes of epitaxial growth with the formation of quantum dots.

In addition, I would slightly change the title of the review and would choose one of two adjectives - either "high-speed" or "ultra-fast".

Thus, this article can be published after minor (but multiple) corrections. In addition, I would recommend polishing the English language with the participation of its native speaker.

Author Response

Reviewer 4:

Review written by Yao et al. is devoted to one of the most actively developing areas of semiconductor physics and technology on the development of ultrafast lasers with quantum dots grown by molecular beam epitaxy. Such lasers are essential for the development of fiber-optic communication. Therefore, the relevance of the review is beyond doubt.

The review is based on the classical scheme, and in its first part, the features of various technologies for the formation of quantum dots (primarily InAs) are considered. The authors then describe in detail the fundamentals of the physics of semiconductor quantum dot (QD) lasers, including various phenomena in nano heterostructures with strong carrier localization that determine a high performance of the QD laser. It should be noted that this chapter is written at a much higher level than the first technological chapter, which is more like an entry-level student essay. Finally, in the final chapter, the authors provide an excellent overview of the state-of-the-art applications of various QD lasers in fiber optic communication devices. In my opinion, this is the most successful and valuable chapter of the review.

To write the review, the authors used a wide range of sources, including both initial and most recent work on quantum dot lasers. I am sure that this review will be interesting and useful to a wide range of readers of the journal, including highly qualified specialists and students.

Most of my comments are technical and are marked in the pdf as comments and suggested changes. The most significant remark concerns the description in the technological chapter (on page 5) of various growth mechanisms:

Response: We appreciate the reviewer's affirmation on our work indeed and suggestions in our paper. The point-to -point responses to the questions can be found below.

Comment 1: I would recommend adding the elastic energy term to the total surface energy (the sum of all energies in the film), which determines the observed epitaxial growth mechanism. This energy depends on the layer thickness and, therefore, determines the transition from a 2D- to 3D growth mechanism in the Stranski-Krastanow mode. In addition, it would be desirable to add the energetic inequalities to Figure 5 and illustrate the change of elastic energy in Figure 7. In my opinion, this addition will raise the level of analysis of surface processes of epitaxial growth with the formation of quantum dots.

Response: Thanks for the reviewer’s instructive suggestions. The descriptions about the energetic inequalities for the Figure 5 have been added into the revised manuscript (Page 7, Figure 5), and the analysis of surface processes of epitaxial growth with the formation of quantum dots was described as follows: “In Phase A, the epitaxial growth follows a 2D growth mechanism in the initial stage of the deposition, the accumulated elastic strain energy increases linearly with the deposited volume, which resulting in a WL on the substrate. When the deposition time reaches tcw, the stable 2D growth is changed to the 2D metastable growth, which means that the epilayers are ready to undergo a transition towards S-K morphology. Phase B can be divided into two steps: the nucleation and growth. The nucleation activation energy is EA-EE, which EE is the excess energy in the metastable 2D layer. When the islands occur, EE decreases and the materials around the islands are consumed for the growth of the islands. There are two factors are concentrated on the nucleation event in a fairly narrow period of time at the point X as shown in Figure 7b. (1) The drop of “super saturation” and (2) the increase of the activation barrier for the thermally activated nucleation” in the revised manuscript. (Page 6, started from line 229).

Comment 2: In addition, I would slightly change the title of the review and would choose one of two adjectives - either "high-speed" or "ultra-fast".

Response: Thanks for the reviewer’s reminding. In this review, two kinds of III-V QD based lasers for optical communication are summarized: one is about the active electrical pumped lasers; and the other is on the passive lasers, instance of semiconductor saturable absorber mirrors mode-locked lasers. Therefore, the inclusion of "high-speed" and "ultra-fast" quantum dot lasers in the title.  

Comment 3: Thus, this article can be published after minor (but multiple) corrections. In addition, I would recommend polishing the English language with the participation of its native speaker.

Response: Thanks to the reviewer for your suggestions and modifications. We have revised all questions included in the attachment, and we have carefully checked and improved the English writing in the revised manuscript.

Round 2

Reviewer 5 Report

The authors state in the Abstract that "In this paper, we review the development of molecular beam epitaxial (MBE) growth methods, material properties, and device characteristics of semiconductor QDs." However, in fact, the characteristics of three groups of injecting lasers obtained by MBE for QDs grown in Stransky-Krastanow mode were compared, plus a group of "passive" lasers with finish coating layers obtained by a different technological method than MBE was added to consideration. Thus, the inclusion of the phrase "development of the MBE" in the title of the paper is rather controversial.

The second phrase in the Abstract really captures the essence of the work presented: "By analyzing cons and pros of different QD lasers from their structures, mechanism and performance, the challenges that arise when using these devices for the applications of fiber-optic communication have been presented". Perhaps the article should be rewritten in this vein.

The part devoted to the types of substrates on which the studied laser structures were grown should be emphasized. At the same time, the objects of discussion of this article were obtained on substrates from various materials and using various technological methods.

The cited literature list contains incorrect citation data. This causes difficulties in revision.

Not all the flaws in the text are eliminated in the article. The text of the manuscript should be more carefully worked over.

Author Response

N/A